# Rbfox1 up-regulation impairs BDNF-dependent hippocampal LTP by dysregulating TrkB isoform expression levels

Francesco Tomassoni-Ardori[1], Gianluca Fulgenzi[1], Jodi Becker[1], Colleen Barrick[1], Mary Ellen Palko[1], Skyler Kuhn[2,3], Vishal Koparde[2,3], Maggie Cam[2,3], Sudhirkumar Yanpallewar[1], Shalini Oberdoerffer[4], Lino Tessarollo[1]*

[1]Neural Development Section, Mouse Cancer Genetics Program, Center for Cancer Research, National Cancer Institute, National Institutes of Health, Frederick, United States; [2]CCR Collaborative Bioinformatics Resource (CCBR), Center for Cancer Research, National Cancer Institute, National Institutes of Health, Frederick, United States; [3]Advanced Biomedical Computational Science, Frederick National Laboratory for Cancer Research sponsored by the National Cancer Institute, Frederick, United States; [4]Laboratory of Receptor Biology and Gene Expression, Center for Cancer Research, National Cancer Institute, National Institutes of Health, Bethesda, United States

**Abstract** Brain-derived neurotrophic factor (BDNF) is a potent modulator of brain synaptic plasticity. Signaling defects caused by dysregulation of its Ntrk2 (TrkB) kinase (TrkB.FL) and truncated receptors (TrkB.T1) have been linked to the pathophysiology of several neurological and neurodegenerative disorders. We found that upregulation of Rbfox1, an RNA binding protein associated with intellectual disability, epilepsy and autism, increases selectively hippocampal TrkB.T1 isoform expression. Physiologically, increased Rbfox1 impairs BDNF-dependent LTP which can be rescued by genetically restoring TrkB.T1 levels. RNA-seq analysis of hippocampi with upregulation of *Rbfox1* in conjunction with the specific increase of TrkB.T1 isoform expression also shows that the genes affected by Rbfox1 gain of function are surprisingly different from those influenced by *Rbfox1* deletion. These findings not only identify TrkB as a major target of Rbfox1 pathophysiology but also suggest that gain or loss of function of Rbfox1 regulate different genetic landscapes.

DOI: https://doi.org/10.7554/eLife.49673.001

*For correspondence:
tessarol@mail.nih.gov

Competing interests: The authors declare that no competing interests exist.

## Introduction

Learning and memory depend on the establishment, maintenance, strengthening or regulated elimination of synapses between neurons. These processes are mediated by neuronal activity and the activity-dependent secretion of factors that act at synapses. BDNF is one of the most potent mediators of synaptic plasticity as it is secreted during Long Term Potentiation (LTP) induction and is functionally essential for acute signaling cascades leading to LTP [reviewed in *Lu et al. (2013)*; *Park and Poo (2013)*. As a consequence, BDNF has a crucial role in cognitive functions (*Bambah-Mukku et al., 2014*; *Panja et al., 2014*). Critical insights into BDNF activities in humans have been provided by the identification of a single-nucleotide polymorphism (SNP) in the *BDNF* gene that converts a valine to methionine at codon 66 (Val66Me) (*Egan et al., 2003*). This polymorphism impairs BDNF trafficking and synaptic localization, causes a reduction in activity-dependent BDNF

secretion and is associated with alterations in brain structure and function leading to several neurological and psychiatric disorders (*Greenberg et al., 2009*; *Lu et al., 2013*). Genetic or pharmacological manipulations of the levels or activity of the BDNF receptor Ntrk2 (TrkB) also result in impaired LTP and reduced synapse numbers causing deficits in the formation and consolidation of hippocampus-dependent memory (*Minichiello, 2009*). TrkB receptor signaling depends on the precise pattern of expression of the different isoforms generated by the *Ntrk2* gene, including the full-length tyrosine kinase receptor (TrkB.FL) and truncated receptors lacking the kinase domain (TrkB.T1). Hypomorphic expression of the TrkB.FL causes hyperphagia-induced obesity due to reduced hypothalamic BDNF signaling, while genetic deletion of TrkB.T1 leads to increased anxiety related behavior associated with structural alterations in neurites of the amygdala (*Carim-Todd et al., 2009*; *Xu et al., 2003*). Moreover, up-regulation of TrkB.T1 levels in brains of a mouse model with a Chromosome 16 trisomia (Ts16) leads to an increased sensitivity of hippocampal neurons to BDNF deprivation due to a block in TrkB.FL functions (*Dorsey et al., 2006*). Importantly, alterations in *NTRK2* isoforms receptor levels have also been associated with neuropsychiatric and neurodegenerative disorders (*Dwivedi et al., 2003*; *Ernst et al., 2009a*; *Ferrer et al., 1999*). Because loss and gain of function experiments have stressed the importance of proper TrkB.T1 expression regulation for normal brain development and function, we elected to investigate this mechanism. We took advantage of the observation that the Ts16 mouse model has a TrkB.T1 upregulation, despite having an intact *Ntrk2* locus on Chromosome 13, to identify genes on Chromosome 16 responsible for the dysregulation of TrkB isoforms expression (*Dorsey et al., 2006*). We identified Rbfox1 as an RNA binding protein that regulates TrkB.T1 receptor levels. Rbfox1 is expressed only in neurons, heart and skeletal muscle, sites with notable TrkB expression. Moreover, *RBFOX1* dysregulation has been associated with intellectual disability, autism, epilepsy and Parkinson disease, pathologies that have been associated with alterations in BDNF signaling as well (*Chao et al., 2006*; *Conboy, 2017*; *Lu et al., 2013*). We found that Rbfox1 upregulation impairs hippocampal BDNF-dependent LTP by specifically increasing TrkB.T1 receptor levels. Although Rbfox1 can regulate the splicing and abundance of many gene isoforms in the nervous system, we show that genetic reduction of the TrkB.T1 isoform in animals with up-regulated Rbfox1 is sufficient to restore hippocampal BDNF-dependent LTP, suggesting that *Ntrk2* is a major target of Rbfox1 pathological dysregulation (*Fogel et al., 2012*; *Gehman et al., 2011*; *Lee et al., 2016*; *Li et al., 2007*; *Underwood et al., 2005*). Importantly, RNA-seq analysis of hippocampi with Rbfox1 upregulation validates the abnormal TrkB.T1 isoform levels, and also shows that the genes affected by increased Rbfox1 levels are different than those changed by its loss of function, thus suggesting that Rbfox1 has broader genetic targets than previously established.

## Results

We have previously reported that the trisomic TS16 mouse model, which has an extra copy of Chromosome (Chr) 16, has a neuronal upregulation of the TrkB.T1 receptor isoform level (*Dorsey et al., 2006*). Since, the *Ntrk2* gene is located in Chr 13 (*Tessarollo et al., 1993*), these data suggested that one or multiple genes present in Chr16 are responsible for this phenotype. A comparison between the genes isolated from brain immunoprecipitation of the spliceosome and bio-informatics analysis of the Ts16 unique region identified two RNA binding proteins, Tra2b (Sfrs10) and Rbfox1 (A2bp1), as potential candidates of TrkB.T1 expression regulation (*Li et al., 2007*). Tra2b is a ubiquitously expressed splicing factor that contributes to alternative splicing processes in a concentration dependent manner (*Elliott et al., 2012*). Therefore, we first tested whether *Tra2b* overexpression in neurons can change TrkB isoforms expression levels. However, infection of primary hippocampal neurons with a *Tra2b*-expressing lentivirus did not cause any change in TrkB isoforms expression (*Figure 1—figure supplement 1*). Next, we tested the RNA binding protein Rbfox1 that, compared to Tra2b, has a more restricted and overlapping pattern of expression with TrkB in organs such as brain, heart and muscle (*Li et al., 2007*; *Stoilov et al., 2002*). Western blot analysis of primary hippocampal neurons transfected with a lentiviral vector expressing *Rbfox1* showed increased truncated TrkB receptor levels associated with Rbfox1 overexpression while the TrkB.FL isoform was unaffected (*Figure 1A,B*). Although TrkB.T1 is the dominant truncated TrkB isoform in mouse brain, the potential presence of other truncated isoforms prompted us to test the identity of the upregulated truncated isoform. As shown in *Figure 1A* (right panel) western blot analysis with an antibody

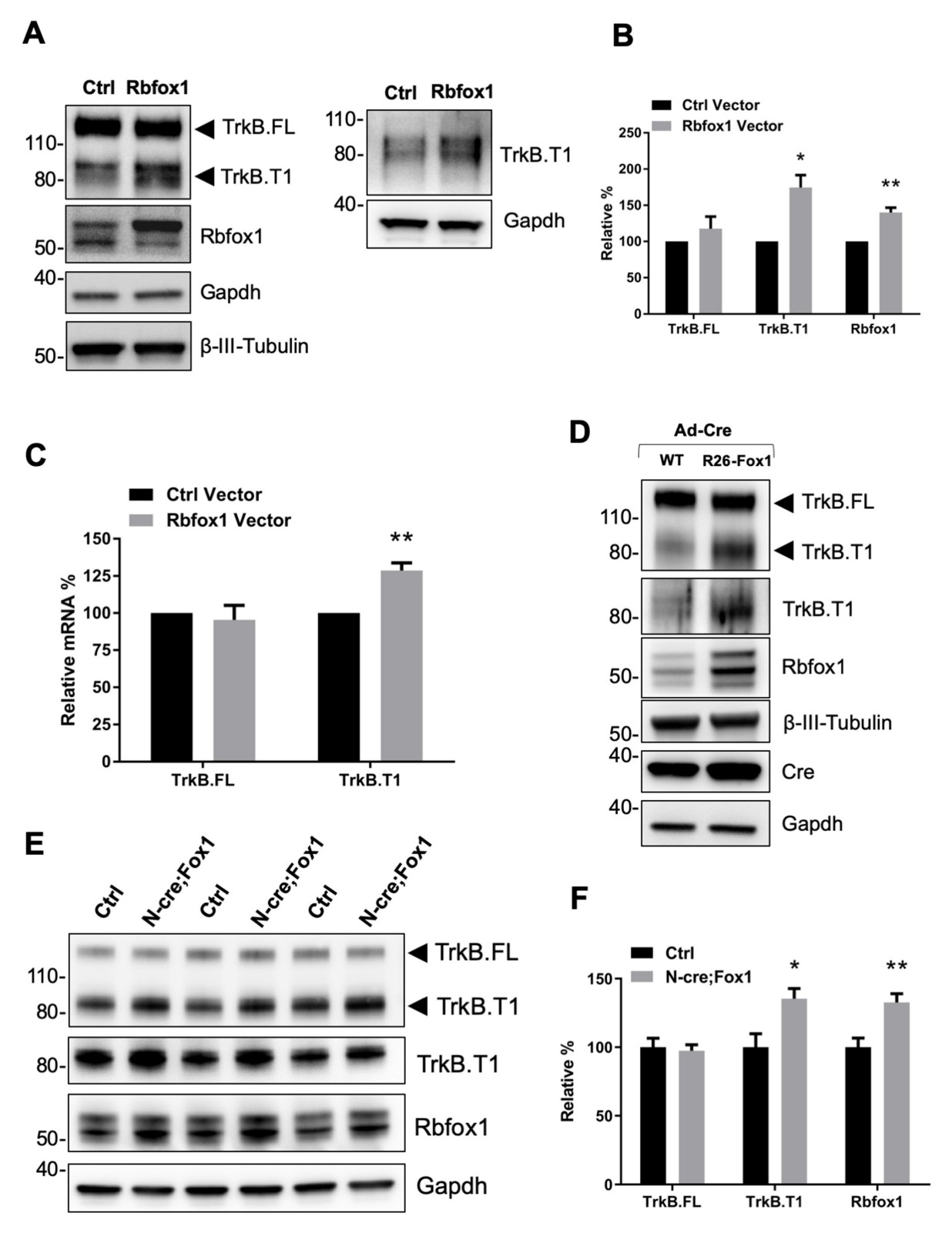

**Figure 1.** *Rbfox1* upregulation increases truncated TrkB. T1 receptor levels in vitro and in vivo. (**A**) Overexpression of Rbfox1 in mouse primary hippocampal neurons leads to a specific up-regulation of TrkB.T1. Western blot analysis of wild-type mouse primary hippocampal neurons transfected with a lentiviral vector expressing *Rbfox1* (Rbfox1) or eGFP (Ctrl) as a control. Neurons were transfected after 4 days in vitro and analyzed 72 hr after transfection. Ntrk2 (TrkB) protein levels were tested with an antibody against the TrkB extracellular domain to detect all TrkB isoforms (left panel) or an

*Figure 1 continued on next page*

*Figure 1 continued*

antibody specifically recognizing TrkB.T1 (right panel). Antibodies against Rbfox1 were used to verify Rbfox1 overexpression; Gapdh and β-III-Tubulin were used as control of loading and as a neuronal marker respectively. (**B**) Quantification of TrkB full-length (TrkB.FL) and TrkB.T1 protein from experiments as in A; n = 3 ± SEM. * = p ≤ 0.05, ** = p ≤ 0.01 (Student's t-test). (**C**) Quantitative PCR analysis of TrkB.FL and TrkB.T1 expression from mouse primary hippocampal neurons as in (**A**); n = 3 ± SEM; ** = p ≤ 0.01 (Student's t-test). (**D**) Immunoblot analysis of littermates wild-type *R26-Rbfox1*$^{+/+}$ (WT) and transgenic *R26-Rbfox1*$^{+/flox}$ (R26-Fox1) primary hippocampal neurons transfected with a Cre-expressing adenovirus (Ad-Cre) and analyzed as in (**A**). (**E**) Immunoblot of hippocampi derived from control *Nes-Cre* animals (Ctrl) and transgenic *Nes-Cre;R26-Rbfox1*$^{+/flox}$ animals (N-cre; Fox1) analyzed with antibodies as in (**A**). (**F**) Immunoblot quantification analysis of TrkB.FL, TrkB.T1 and Rbfox1 protein levels of N-cre;Fox1 relative to Ctrl as in (**E**); n = 6 ± SEM; * = p ≤ 0.05; ** = p ≤ 0.01 (Student's t-test).

DOI: https://doi.org/10.7554/eLife.49673.002

The following figure supplements are available for figure 1:

**Figure supplement 1.** Overexpression of Tra2b does not change TrkB receptor expression.
DOI: https://doi.org/10.7554/eLife.49673.003
**Figure supplement 2.** Generation and characterization of a mouse model with inducible *Rbfox1* expression.
DOI: https://doi.org/10.7554/eLife.49673.004
**Figure supplement 3.** Expression of the TrkC receptor isoforms and Rbfox2 is not affected by Rbfox1 upregulation.
DOI: https://doi.org/10.7554/eLife.49673.005

recognizing the specific 11 amino acid intracellular domain of TrkB.T1 confirmed that TrkB.T1 is the isoform upregulated by Rbfox1. Moreover, QPCR analysis also revealed an increased level of TrkB.T1-specific mRNA while TrkB.FL was unchanged, paralleling the results from the western analysis (*Figure 1C*).

To study the physiological significance of TrkB.T1 upregulation by Rbfox1 we generated a mouse model with an inducible gain of function *Rbfox1* allele. By gene targeting of the *Gt(ROSA)26Sor* locus we introduced one copy of the *Rbfox1* cDNA preceded by the CAG promoter to drive expression (R26-Fox1; *Sakai and Miyazaki, 1997*). A loxP-flanked stop cassette was inserted between the CAG promoter and the *Rbfox1* c-DNA to allow for conditional activation of the gene (*Figure 1—figure supplement 2A*). Primary hippocampal neurons from heterozygous *Rbfox1* transgenic mice (R26-Fox1) were isolated and transduced with a *Cre*-recombinase-expressing adenovirus (Ad-Cre) to delete the stop cassette and induce *Rbfox1* expression. Compared to control neurons infected with the Ad-Cre, R26-Fox1 neurons showed increased Rbfox1 expression and subsequent TrkB.T1 upregulation (*Figure 1D*). These data suggest that this is an efficient model to up-regulate Rbfox1 and further validated our initial observation that lentiviral overexpression of Rbfox1 upregulates TrkB.T1 (*Figure 1A–C*).

Next, we tested whether Rbfox1 upregulation can regulate TrkB.T1 expression in vivo by crossing the R26-Fox1 model with the *Nes-cre* (N-cre) mouse (*Tronche et al., 1999*). First we tested *Nes-cre* expression in the hippocampus by crossing it with a *Gt(ROSA)26Sor-LacZ* (R26-LacZ) reporter mouse. β-Galactosidase staining (X-Gal) of the hippocampal area showed LacZ activity in granule cells of dentate gyrus (DG) and pyramidal cells of CA1 and CA3 regions, as previously reported (*Chen et al., 2016*; *Schultz et al., 2011*) (*Figure 1—figure supplement 2B*). Importantly, *Nes-Cre* (N-cre) crossing with the R26-Fox1 transgenic mouse (N-cre;Fox1) showed that hippocampal Rbfox1 expression is similar between N-cre and N-cre;Fox1 mice, and Rbfox1 expression does not co-localize with the glial marker, glial fibrillary acidic protein (GFAP) (*Figure 1—figure supplement 2C,D*). These data demonstrate the relevance of this model to study Rbfox1 neuronal up-regulation in vivo. N-cre;Fox1 animals showed a modest 25% increase of *Rbfox1* mRNA level in the hippocampus (data not shown). At the protein level, similar to what observed in vitro, N-cre;Fox1 hippocampi had a significant 35% increase in TrkB.T1 associated with about 33% Rbfox1 protein upregulation, while TrkB.FL level was unaffected (*Figure 1E,F*). mRNA analysis also showed that TrkB.T1, but not TrkB.FL was significantly increased as a consequence of Rbfox1 upregulation (data not shown). Importantly, this model leads to a relatively modest up-regulation of Rbfox1 expression suggesting its validity to study the physiological consequences of Rbfox1 dysregulation.

To further test the specificity of this phenotype, we also investigated the protein levels of Ntrk3 (TrkC), another *Ntrk* family member also expressed in hippocampal neurons (*Tessarollo et al., 1993*), in the same N-cre;Fox1 hippocampi and found that the levels of TrkC isoform receptors were unaffected by in vivo Rbfox1 upregulation (*Figure 1—figure supplement 3*).

Since Rbfox1 upregulation increases TrkB.T1 expression in vivo (*Figure 1E,F*) we also tested whether lower levels of Rbfox1 cause a reduction of TrkB.T1 expression. Western blot analysis of hippocampi dissected from *Rbfox1* heterozygous (*Gehman et al., 2011*) animals showed a significant 50% reduction in Rbfox1 protein levels. However, there was no change in TrkB.T1 or TrkB.FL receptor isoforms expression (*Figure 2A,B*). Therefore, we tested whether other *Rbfox* family members could compensate for the partial loss of *Rbfox1*. Hippocampal lysates from *Rbfox1* heterozygous animals showed a significant 35% upregulation of Rbfox2 whereas Rbfox3 expression was unchanged [*Figure 2A,B*; (*Vuong et al., 2018*). This result strongly suggests that *Rbfox1* is haploinsufficient. Moreover, it suggests that Rbfox2 is upregulated by Rbfox1 reduction providing a possible explanation for why TrkB receptor isoform expression is not altered in the hippocampus of *Rbfox1* heterozygous mice. To further test compensatory mechanisms by other *Rbfox* genes, we overexpressed *Rbfox2* and *Rbfox3* in primary hippocampal neurons by lentiviral transduction. Interestingly, overexpression of Rbfox2 but not Rbfox3 caused TrkB.T1 upregulation as observed in N-cre-Rbfox1 transgenic mice (*Figure 2C,D*) suggesting functional redundancy at least in the context of TrkB.T1 receptor regulation. However, contrary to the upregulation of Rbfox2 caused by loss of one *Rbfox1* allele, Rbfox2 expression was not downregulated in response to increased Rbfox1 expression (*Figure 1—figure supplement 3*).

The finding that overexpression of Rbfox1 causes a specific increase in the amount of TrkB.T1 mRNA without affecting the full-length receptor (TrkB.FL) suggests that Rbfox1 may act by increasing TrkB.T1 mRNA stability or expression and not as an alternative splicing factor. Therefore, we first analyzed Rbfox1 iCLIP datasets in brain to investigate the presence of *Ntrk2* transcripts among the Rbfox1 targets (*Figure 3A*; *Damianov et al., 2016*). iCLIP experiments were performed from two distinct mouse brain nuclear fractions including a high molecular weight fraction (HMW) which is more intimately associated with chromatin and enriched in pre-mRNA species, and a more soluble nuclear fraction which excludes chromatin associated protein like histone H3 and is enriched in mature mRNAs (*Damianov et al., 2016*). Analysis of both HMW and soluble nuclear fractions showed several iCLIP-hits along the *Ntrk2* sequence covering both intronic as well as 3'-UTR regions of the gene, suggesting direct binding of Rbfox1 to *Ntrk2* transcripts (TrkB.FL and TrkB.T1) (*Figure 3A*).

To further validate Rbfox1 binding to *Ntrk2* mRNAs we performed RNA immunoprecipitation (RIP) experiments (*Figure 3C*) (*Jayaseelan et al., 2011*). Lysates from wild type hippocampal neurons were subjected to immunoprecipitation (IP) with a Rbfox1 specific monoclonal antibody (1D10, a kind gift of Dr. Douglas Black) followed by RT-PCR analysis to detect bound RNA. While immunoprecipitation with control IgG failed to pull down any of the tested mRNA, IP with the anti-Rbfox1 antibody pulled down both the TrkB.FL and TrkB.T1 mRNA as well as a proximal intronic region upstream of the specific TrkB.T1 exon (*Figure 3C*), confirming the association found in the iCLIP experiments. Calmodulin-binding transcription activator 1 (*Camta1*) mRNA, used as positive control (*Gehman et al., 2011*; *Lee et al., 2016*), was also pulled down whereas Sirtuin 1 (*Sirt1*), used as a negative control, was not, even though it is present in the neuron lysates like all the other mRNAs (*Figure 3C*, Input lanes).

The iCLIP analysis also showed the presence of hit-clusters in the TrkB.T1 3'-UTR region of both TrkB.FL and TrkB.T1 (*Figure 3A*) suggesting a possible role for Rbfox1 in stabilizing TrkB mRNAs by antagonizing specific miRNAs binding to the 3' UTR (*Lee et al., 2016*). Therefore, we generated a HEK293 cell line with doxycycline-inducible *Rbfox1* expression and transfected it with TrkB.T1 cDNA containing the 3'UTR sequence (*Figure 3—figure supplement 1A,B*). Surprisingly, despite the presence in the 3'UTR of six 'GCATG' Rbfox1 binding sites, expression of Rbfox1 caused no change in the level of expression of TrkB.T1 suggesting that the TrkB.T1 mRNA isoform is regulated by Rbfox1 through a different mechanism not involving the 3'UTR region. Similar experiments done in a mouse neuroblastoma cell line (Neuro-2a) confirmed that this result is independent of the cell type (*Figure 3—figure supplement 1C,D*).

Since Rbfox1 binds both TrkB.T1 and TrkB.FL mRNAs, we next tested whether Rbfox1 upregulation differentially regulates the stability of the two mRNA isoforms by using ethynyl-uridine (EU) to label new nascent RNA followed by purification and QPCR analysis. Measurements of newly synthesized *Ntrk2* mRNAs levels with basal or up-regulated expression of Rbfox1 showed that TrkB.T1, but not TrkB.FL mRNA was significantly stabilized (p = 0.03) in primary hippocampal neurons by the increased levels of Rbfox1 (*Figure 3D*). Testing of the same samples for *Ntrk3* (TrkC) mRNA

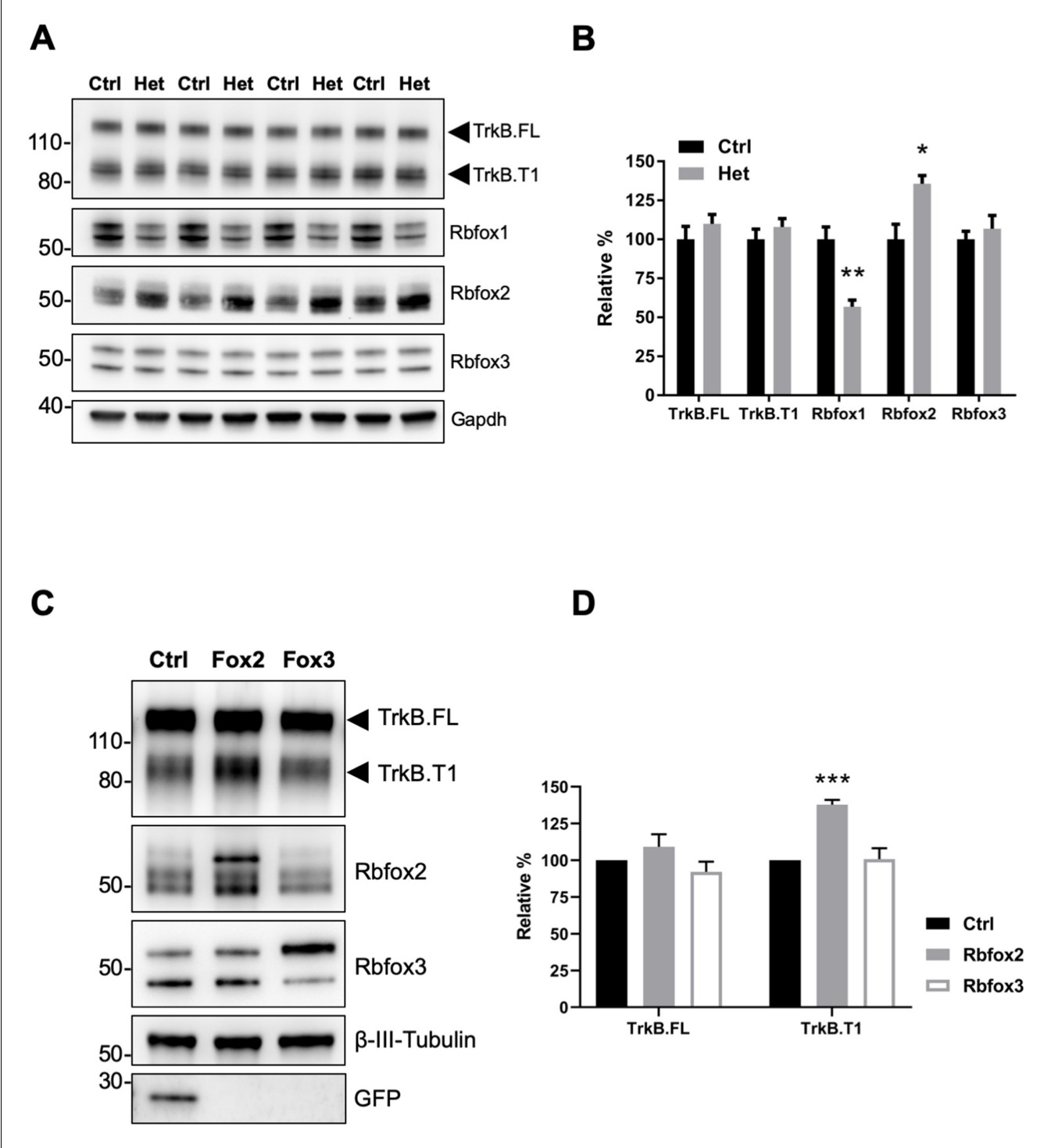

**Figure 2.** Rbfox2 compensates for *Rbfox1* heterozygosity in vivo and its overexpression increases TrkB. T1 receptor levels. (A) Western blot analysis of control (Ctrl) and *Rbfox1*$^{+/-}$ (Het) mouse hippocampus with antibodies against TrkB, Rbfox1, Rbfox2, Rbfox3 and Gapdh (used as loading control). (B) quantifications of immunoblots bands from (A) relative to Ctr (100%); n = 4 ± SEM; *$p \leq 0.05$; **$p \leq 0.01$ (Student's t-test). (C) Overexpression of Rbfox2, but not Rbfox3 up-regulate TrkB.T1 expression in primary hippocampal neurons. Western blot analysis of wild-type mouse primary hippocampal neurons transfected with a lentiviral vector expressing *eGFP* as a control (Ctrl), *Rbfox2* (Fox2) or *Rbfox3* (Fox3). Neurons were transfected after 4 days in vitro and analyzed 72 hr after transfection by immunoblot with antibodies against TrkB extracellular domain to detect all TrkB isoforms, GFP, Rbfox2, Rbfox3 and β-III-Tubulin as loading control and neuronal marker. (D) Quantification of TrkB full-length (TrkB.FL) and TrkB.T1 protein from experiments as in (C); n = 3 ± SEM. *** = $p \leq 0.001$ (Student's t-test).

DOI: https://doi.org/10.7554/eLife.49673.006

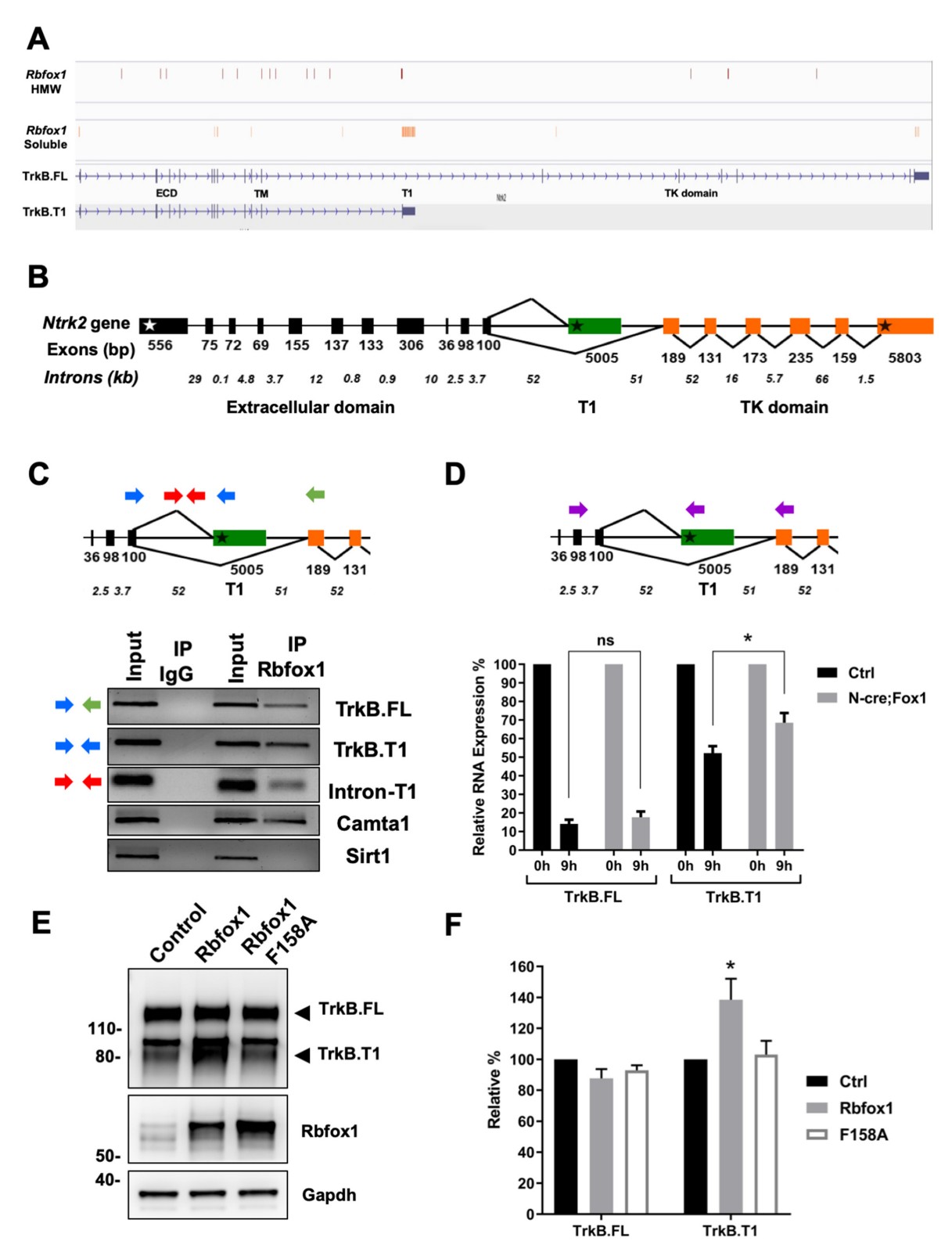

**Figure 3.** Rbfox1 upregulation increases mRNA stability of the TrkB. T1-isoform and its RNA binding function is necessary to increase TrkB.T1 receptor levels. (A) Rbfox1 iCLIP analysis of mouse brain chromatin associated high molecular weight nuclear fraction (HMW) or soluble nuclear fraction (Soluble) (*Damianov et al., 2016*) showing association of Rbfox1 to TrkB transcripts in both HMW (vertical red bars) and soluble (vertical orange bars) nuclear fractions. Extracellular domain (ECD), specific TrkB.T1 region (**T1**) and tyrosine kinase domain (TK domain) are also indicated. (B) Schematic

*Figure 3 continued on next page*

*Figure 3 continued*

representation of the murine *Ntrk2* gene. Exon lengths are indicated in base pairs (bp) while intron lengths are indicated as kilo-bases (kb). White and black stars indicate the start and stop codons respectively. (C) Agarose gels of RT-PCR amplification products from an RNA immunoprecipitation (RIP) analysis of wild type primary hippocampal neurons with an Rbfox1 antibody; input samples are from the PCR amplification of the RNA and antibody mixture before the immunoprecipitation (IP) with mouse IgG or Rbfox1 antibodies. Indicated exonic (blue and green arrows) and intronic (red arrows) primers were used for the analysis. Note the presence of RT-PCR amplification bands following Rbfox1 IP for both TrkB.FL and TrkB.T1 and a proximal intronic region upstream of the specific TrkB.T1 exon, *Camta1* (positive control; *Gehman et al., 2011*; *Lee et al., 2016*), but not *Sirt1* (negative control). (D) Pulse-chase mRNA stability assay in primary hippocampal neurons derived from littermate control *R26-Rbfox1*[+/flox] (Ctrl) and *Nes*-Cre;*R26-Rbfox1*[+/flox] (N-cre;Fox1) embryos. Nascent RNA of primary neurons was labeled with 5-EU for 5 hr (pulsing) followed by QPCR analysis at 0 and 9 hr using c-DNA specific primers (purple arrows in schematic) for TrkB.FL and TrkB.T1. Values at 9 hr are expressed as percentage relative to 0 hr. $n = 6 \pm$ SEM; ns = $p > 0.05$; * = $p \leq 0.05$ (Student's t-test). (E); Rbfox1 RNA binding activity is required to promote TrkB.T1 up-regulation. Western blot analysis of wild-type mouse primary hippocampal neurons transfected with an adenovirus expressing WT *Rbfox1* (Rbfox1) or an *Rbfox1* with a mutation in the RNA binding domain (Rbfox1-F158A). Neurons were transfected after 4 days in vitro and analyzed 48 hr after transfection. (F) Quantification of TrkB.FL and TrkB.T1 protein from experiments as in E; $n = 3 \pm$ SEM. * = $p \leq 0.05$, (Student's t-test).

DOI: https://doi.org/10.7554/eLife.49673.007

The following figure supplements are available for figure 3:

**Figure supplement 1.** The TrkB.

DOI: https://doi.org/10.7554/eLife.49673.008

**Figure supplement 2.** Rbfox1 upregulation does not change TrkC isoforms RNA stability.

DOI: https://doi.org/10.7554/eLife.49673.009

transcripts, showed that neither TrkC.FL nor the truncated TrkC.T1 mRNA stability was affected by Rbfox1 upregulation (*Figure 3—figure supplement 2*). These data suggest a specific role of this RBP in stabilizing the TrkB.T1 mRNA receptor isoform.

We next asked whether Rbfox1 RNA binding activity is required for increasing TrkB.T1 levels. Adenoviruses containing a wild type or a *Rbfox1* mutant lacking RNA binding capability (F158A) were used to transduce primary hippocampal neurons (*Hakim et al., 2010*; *Jin et al., 2003*). As shown in *Figure 3E–F*, overexpression of Rbfox1-mutant failed to increase TrkB.T1 levels suggesting that Rbfox1 RNA binding activity is essential for this function.

The findings that loss of function of Rbfox1 causes an upregulation of Rbfox2 and does not influence TrkB.T1 expression (*Figure 2A,B*) whereas Rbfox1 gain of function does not change Rbfox2 levels and increases TrkB.T1 expression (*Figure 1—figure supplement 3*) suggest that up or down regulation of Rbfox1 may target different genes. To test this hypothesis, we performed whole transcriptome deep coverage RNA-seq analysis of hippocampi (with an average of 100 million reads) from N-cre;Fox1 and control N-cre animals. The RNA-seq confirmed our findings that *Rbfox2* mRNA is not modulated in N-cre;Fox1 transgenic animals (*Supplementary file 1*). More importantly, the RNA-seq analysis confirmed that TrkB.T1 mRNA was the only upregulated *Ntrk2* isoform (*Figure 4*). It also showed that among all the *Ntrk2* isoforms annotated in the Ensembl database the TrkB.T1 (*Ntrk2-202*) and TrkB.FL (*Ntrk2-201*) isoforms are by far the most highly expressed in the hippocampus with approximately 15,000 and 12,000 reads respectively, whereas the other isoforms combined represent only about 5% of the total TrkB transcripts (*Figure 4B*). In western blot analysis, an almost complete lack of signal at the level of the truncated isoform in the hippocampus of TrkB.T1 knockout mice also confirmed that TrkB.T1 is the main truncated isoform expressed in this region and no other truncated isoforms are present (*Figure 4C*).

Gene ontology (GO) term analysis of the RNA-seq data revealed that the major biological processes affected in the N-cre;Fox1 hippocampus are involved in important neuronal functions comprising synapse organization, neurotransmitter secretion, axonal development and many others (*Figure 5A*; *Supplementary file 1*). Interestingly, we found that the TrkB-receptor ligand *Bdnf* is upregulated in the N-cre;Fox1 mouse although it is not clear whether this is the result of a compensatory mechanism in response to the upregulation of TrkB.T1 or whether *Bdnf* is a direct target of Rbfox1 (*Supplementary file 1*). The most surprising result came from the parallel analysis, using identical parameters, of the *Rbfox1*-KO and the N-cre;Fox1 hippocampal RNA-seq raw data (*Vuong et al., 2018*). In fact, we found that the gene-isoforms that are differentially expressed in the hippocampus of N-cre;Fox1 mice are mostly different from the gene-isoforms affected by *Rbfox1* deletion [*Figure 5B*; *Supplementary file 1* and *2*; (*Vuong et al., 2018*) (*Gehman et al., 2011*).

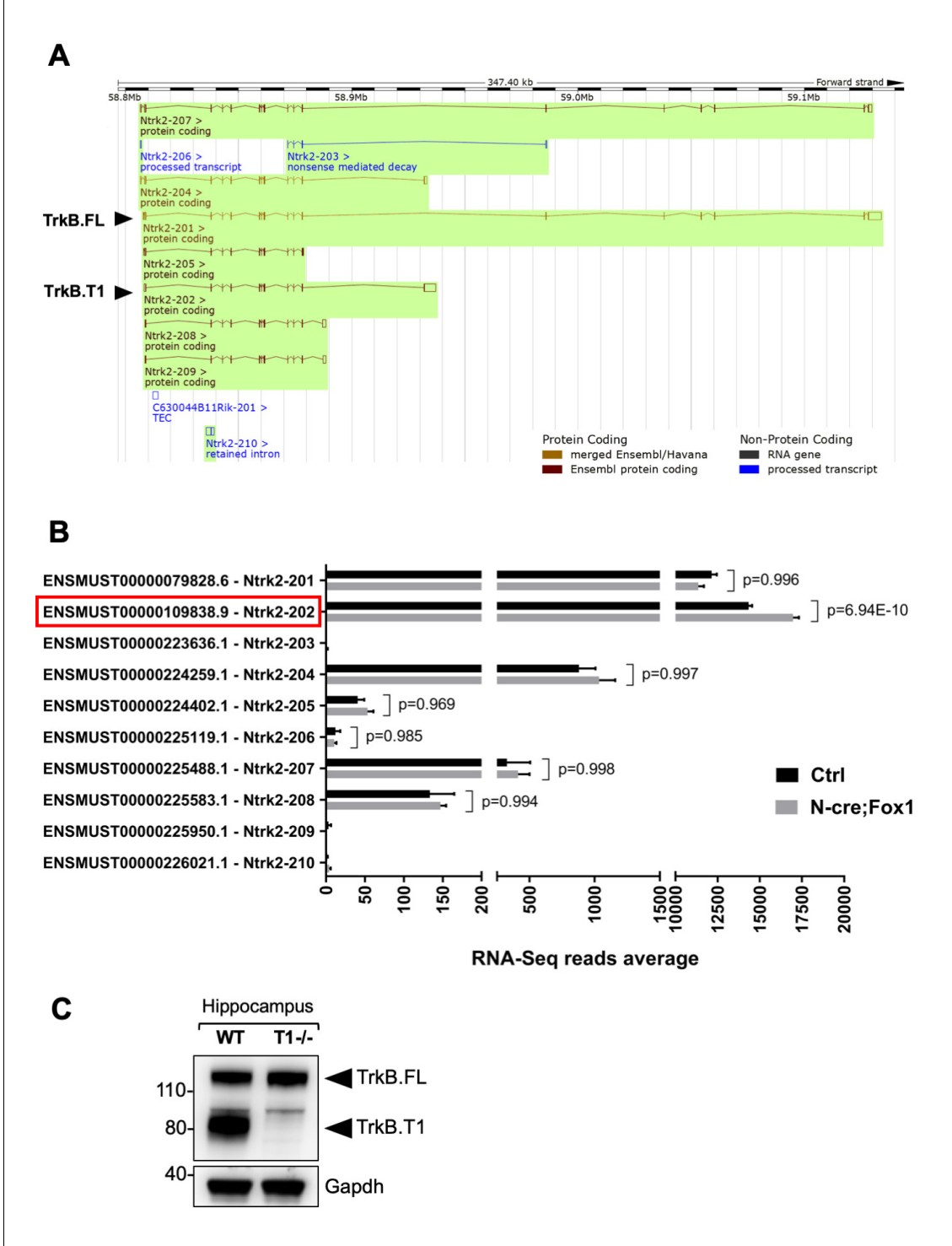

**Figure 4.** RNA-Seq deep coverage analysis of *Ntrk2* (TrkB) transcripts shows specific increase of the TrkB. T1 isoform levels in the hippocampus of Rbfox1 overexpressing animals. (**A**) Schematic representation of the murine *Ntrk2* (TrkB) gene transcripts annotated from ENSEMBL (highlighted in green). In each transcript exons and introns are indicated with vertical and horizontal lines respectively. TrkB.FL (Ntrk2-201) and TrkB.T1 (Ntrk2-202) are indicated (Picture modified from Ensembl genome browser; ENSMUSG00000055254). (**B**) RNA-Seq specific analysis of the *Ntrk2* transcripts in control (Ctrl; *Nes-Cre*) and N-cre;Fox1 hippocampus (*Nes-Cre;R26-Rbfox1*[+/flox]). Isoform IDs are indicated on the Y axis while the average of RNA-sequencing reads for every transcript is indicated on the X axis. TrkB.T1 (Ntrk2-202; red box) is the only TrkB isoform significantly modulated in the N-cre;Fox1 hippocampus. (**C**) Immunoblot analysis of wild type (WT) and TrkB.T1 knockout (T1-/-) hippocampus blotted with an antibody against the TrkB extracellular domain.

DOI: https://doi.org/10.7554/eLife.49673.010

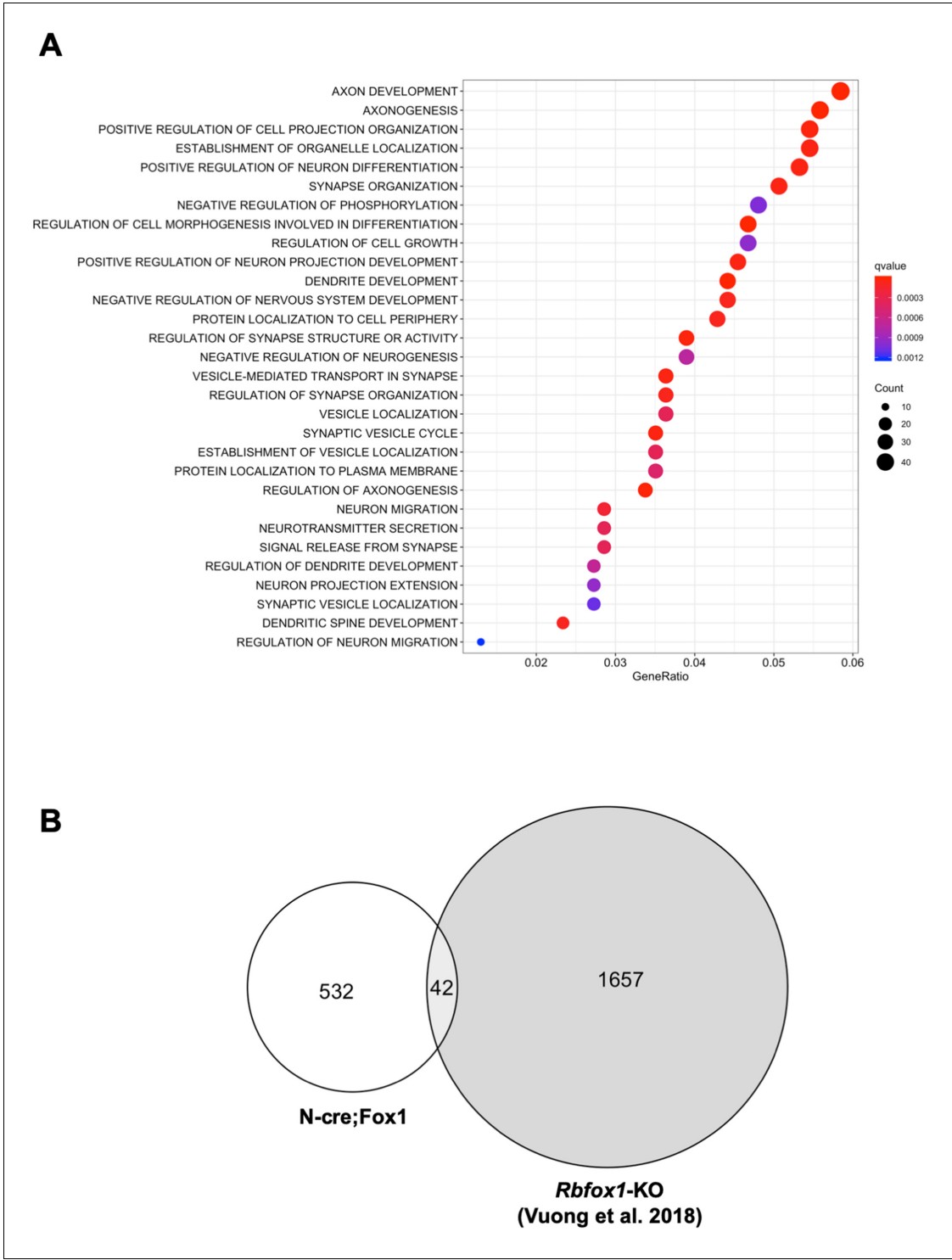

**Figure 5.** Upregulation or downregulation of *Rbfox1* changes the expression of different gene-isoforms. (**A**) Dot plot showing the top enriched biological process Gene Ontology (GO) terms. Note that significant over-represented GO terms (q-value <0.05) are neuronal related processes. GO terms are displayed on the left Y axis. (**B**) Venn diagram showing the number of overlapping gene-isoforms differentially expressed in N-cre;Fox1 and *Rbfox1* knockout (*Rbfox1*-KO) hippocampus (*Vuong et al., 2018*).

DOI: https://doi.org/10.7554/eLife.49673.011

The following figure supplement is available for figure 5:

**Figure supplement 1.** Genes modulated by *Rbfox1* upregulation are largely un-affected by *Rbfox1* knockout.
DOI: https://doi.org/10.7554/eLife.49673.012

These findings were further supported by the very limited or complete lack of overlap of genes that are dysregulated by hippocampal *Rbfox1* gain of function (N-cre;Fox1) or *Rbfox1* loss of function in brain (*Rbfox1*-KO) (*Gehman et al., 2011*) or hippocampal neurons lacking both *Rbfox1* and *Rbfox3* [(*Figure 5—figure supplement 1*) (*Lee et al., 2016*).

Next, we decided to investigate the physiological consequences of TrkB.T1 changes of expression. Rbfox1 upregulation causes a specific increase in the levels of TrkB.T1 without changing the expression of TrkB full-length (TrkB.FL). Therefore, we created an in vitro system to mimic this situation and study BDNF/TrkB signaling. We first generated a cell line with stable expression of TrkB.FL (HEK293-TrkB.FL) followed by transfection with a plasmid to express increasing amounts of TrkB.T1. Stimulation with BDNF showed an inverse correlation between TrkB.T1 expression levels and TrkB. FL (p-TrkB.FL) and ERK (p-ERK) phosphorylation suggesting an impairment in TrkB.FL signaling when TrkB.T1 is upregulated (*Figure 6A*). Importantly, deletion of TrkB.T1 in primary hippocampal neurons increased both basal as well as BDNF-stimulated p-TrkB.FL and p-ERK compared to control neurons suggesting that TrkB.T1 expression levels are potent regulators of TrkB.FL signaling (*Figure 6B,C,D*).

Since BDNF and TrkB are potent modulators of synaptic transmission and plasticity, we then tested whether TrkB.T1 upregulation would affect these brain activities [reviewed in *Lu et al. (2013)*. Although we showed that TrkB.T1 is up-regulated in the hippocampus of N-cre;Fox1 mice, we investigated TrkB expression at the synaptic terminals by analyzing hippocampal synaptosomes (*Figure 7*). Immunoblot analysis of hippocampal synaptosomal fractions from controls and N-cre;Fox1 mice showed that Rbfox1 leads to a significant ~20% upregulation in TrkB.T1 but not TrkB.FL protein levels at the synaptic terminals. The synaptic protein PSD-95 used as control was enriched in the synaptosomal fraction but unaffected by Rbfox1 overexpression (*Figure 7A,B*).

Therefore, we tested the physiological significance and net outcome of Rbfox1-induced upregulation of TrkB.T1 by studying hippocampal synaptic plasticity (*Figure 8*). Due to the fact that Rbfox1 affects several genes regulating synaptic plasticity [*Figure 5A*; *Supplementary file 1*; (*Gehman et al., 2011*) we focused on BDNF-induced long term potentiation (LTP) at the Schaffer collateral-CA1 neuron synapses to dissect the specific effect on LTP caused by alterations in BDNF/TrkB signaling. We applied extracellular BDNF (20 ng/ml) to hippocampal slices from control mice for 20 min while the field excitatory postsynaptic potentials (fEPSPs) were elicited at 20 s intervals. As previously reported, BDNF induced a strong, gradual increase in the fEPSP in control N-cre (Ctrl) hippocampi (*Kang and Schuman, 1995*). Surprisingly, N-cre;Fox1 animals had a blunted induction of LTP in response to BDNF (*Figure 8*) and at 50 min this response was only about 30% of Ctrl (Ctrl 218.14 ± 22.42% vs. N-cre;Fox1 72.95 ± 12.82%) (*Figure 8A–C*). To test whether the BDNF-induced LTP deficit in the mutant animals was caused by the augmented TrkB.T1 expression, we introduced a TrkB.T1 KO allele in N-cre;Fox1 (N-cre;Fox1;T1+/-) mouse model to reduce its levels. As shown in *Figure 8—figure supplement 1*, this strategy led to a significant, although partial rescue of the BDNF-induced LTP deficit caused by Rbfox1 upregulation. Interestingly, analysis of BDNF-induced LTP in TrkB.T1 heterozygous animals (T1+/-) was similar to that of controls, suggesting that TrkB.T1 is not haploinsufficient and its levels influence BDNF-induced LTP only in the situation of upregulation (*Figure 8—figure supplement 2*). To further test whether TrkB.T1 levels are most critical in neurons, where Rbfox1 is expressed, we introduced a conditional TrkB.T1-loxP allele into the N-cre; Fox1 model (N-cre;Fox1;T1-Flx). This strategy allows for the simultaneous induction of *Rbfox1* activation, which causes TrkB.T1 upregulation, and deletion of one copy of TrkB.T1 in the same neurons where *Nes-cre* is active. Importantly, this strategy also led to a partial rescue of BDNF-induced LTP over time, which became significant at 50 min (187.97 ± 49.26%) strongly suggesting that the TrkB. T1 up-regulation in neurons caused by Rbfox1 is causing the impairment in BDNF-induced LTP (*Figure 8A–C*). Analysis of the Shaffer collateral-CA1 excitability (Input/Output relationship *Figure 8E*) and presynaptic function with paired pulse stimulation (*Figure 8F*) showed no differences between the controls and mutant animals suggesting that hippocampal circuitries overall are not affected by the genetic manipulations. Taken together, these data strongly suggest that Rbfox1 upregulation leads to a deficit in the BDNF-induced LTP caused, at least in part, by an increase in TrkB.T1 receptor expression.

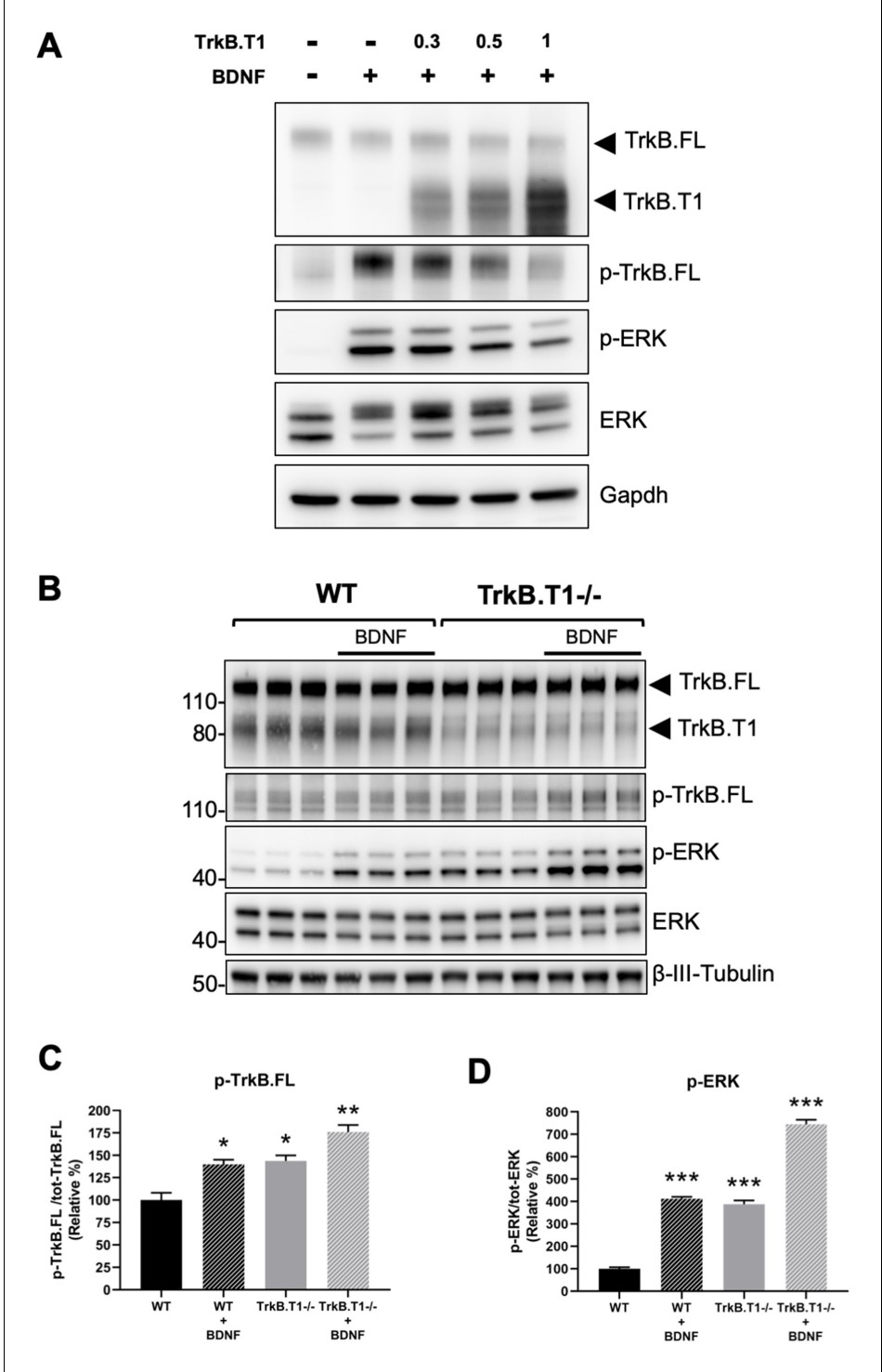

**Figure 6.** TrkB.T1 receptor expression regulates TrkB.FL signaling. (A) TrkB activation by BDNF is decreased by increased levels of TrkB.T1. HEK293 cells stably expressing TrkB.FL were transfected with increasing amounts of a TrkB.T1 expressing plasmid (0.3 μg, 0.5 μg and 1 μg) before BDNF treatment (5 ng/ml for 5 min). Immunoblots were probed with antibodies against TrkB extracellular domain to detect both TrkB.FL and TrkB.T1, phospho-TrkB.FL at tyrosine 515 (p-TrkB.FL), phospho-ERK (p-ERK), ERK and Gapdh as controls. (B) TrkB.T1- /- neurons have increased BDNF signaling. Immunoblot analysis of E18.5 WT and TrkB.T1- /- primary hippocampal neurons cultured for 6 days in vitro before BDNF treatment (1 ng/ml for 5 min; black bar). Antibodies are as in (A) except for the β-III-tubulin antibody used as neuronal marker. (C, D) Immunoblot quantification analysis from (B) of p-TrkB.FL (C) and p-ERK (D) expressed respectively, as relative percentage of p-TrkB.FL over total TrkB.FL and phospho-ERK over total ERK. n = 3 ± SEM. * = p ≤ 0.05, ** = p ≤ 0.01, *** = p ≤ 0.001 (Student's t-test).
DOI: https://doi.org/10.7554/eLife.49673.013

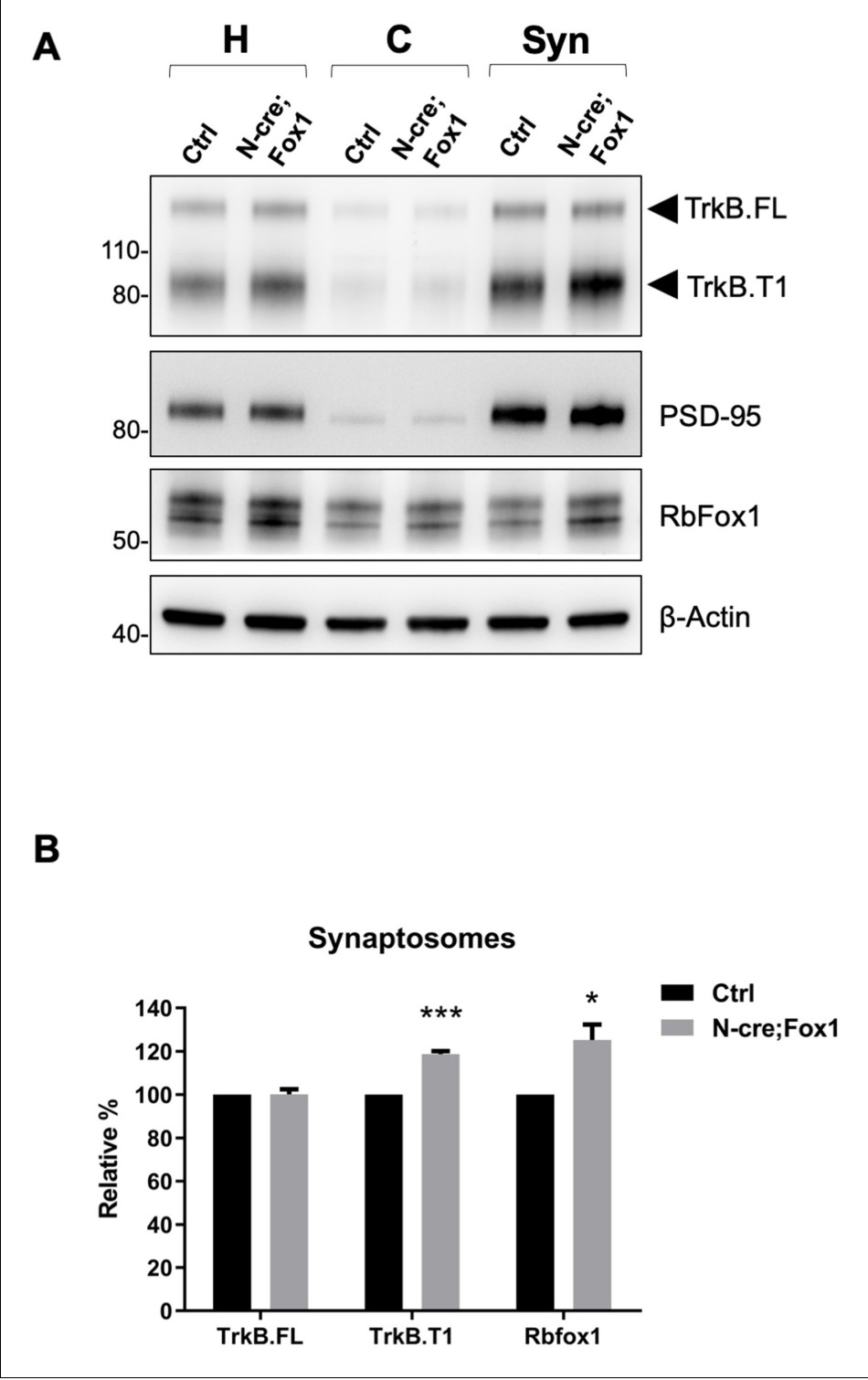

**Figure 7.** Upregulation of Rbfox1 leads to increased truncated TrkB. T1 in synaptosomes. (**A**) Western blot analysis of TrkB isoform, PSD-95 and Rb-Fox1 expression in total homogenates (H), cytosolic (C) and synaptosome fraction (Syn) from hippocampus of control (Ctrl; *Nes*-Cre) and N-cre;Fox1 mouse brains. PSD-95 was used as control for enrichment of the synaptosomal fraction. (**B**) Immunoblot quantification of TrkB.FL, TrkB.T1 and Rbfox1 specific bands in the synaptosomal fraction as in (**A**); n = 3 ± SEM, *** = p ≤ 0.001, * = p ≤ 0.05 (Student's t-test).
DOI: https://doi.org/10.7554/eLife.49673.014

# Discussion

Genomic mutations affecting *RBFOX1* expression have been associated with intellectual disability, epilepsy, autism and Parkinson's disease (*Bill et al., 2013*; *Conboy, 2017*; *Lin et al., 2016*). Rbfox1 is an RBP that regulates the RNA metabolism of hundreds of genes expressed in neurons; thus, a major challenge has been to identify those genes that are in a nodal position downstream of Rbfox1 in transducing its normal and pathological functions (*Gehman et al., 2011*; *Lee et al., 2016*; *Vuong et al., 2018*; *Weyn-Vanhentenryck et al., 2014*). So far, the recognized experimental paradigms to identify and study genes targeted by Rbfox1 have been by downregulation or knockout experiments (*Gehman et al., 2011*; *Lee et al., 2016*). Here we show that *Rbfox1* upregulation influences the neuronal expression level of the BDNF receptor TrkB.T1, a change not present in *Rbfox1* deletion experiments, and this regulation has significant physiological consequences. Importantly, whole transcriptome RNA-seq analysis of hippocampi from mice with *Rbfox1* upregulation revealed a completely different set of differentially expressed gene-isoforms when compared to those identified by *Rbfox1* knock-out experiments. Although there are limitations associated with comparing data sets from different studies our conclusions are supported by the parallel analysis of raw data from all studies (*Vuong et al., 2018*) using uniform parameters and filtering, and similar findings from other independent models of *Rbfox1* loss of function (*Figure 5—figure supplement 1*; *Lee et al., 2016*; *Gehman et al., 2011*).

These results suggest that upregulation or downregulation of a specific RBP can have different genetic outcomes and may be relevant to understanding how to approach pathologies caused by RBP dysregulation. A notable example is provided by *Tra2b* (*Sfrs10*), an RBP ubiquitously expressed but whose differences in cellular concentrations is believed to lead to tissue-specific pattern of splicing (*Elliott et al., 2012*). Moreover, in spinal muscular atrophy (SMA), a disease caused by deletion of the *SMN1* gene, expression of the adjacent full-length *SMN2* gene, that includes exon7, can ameliorate disease in some patients. Tra2b overexpression in transfected cells appears to increase splicing of *SMN2* exon seven promoting the production of a full-length SMN2, thus suggesting that this strategy could ameliorate SMA disease (*Hofmann et al., 2000*). However, *Tra2b* deletion in mice (*Sfrs10$^{-/-}$*) does not influence splicing of *SMN2* exon 7 (*Mende et al., 2010*).

Upregulation of Rbfox1 causes increased expression of TrkB.T1 which in turn blunts BDNF-induced LTP. RNA-seq analysis, besides validating TrkB.T1 up-regulation, identified more than five hundred differentially expressed gene-isoforms in the hippocampus of N-cre;Fox1 animals, including many involved in important neuronal functions (*Figure 5*; *Supplementary file 1*). However, decreasing only TrkB.T1 levels in the same cells where Rbfox1 is up-regulated, is sufficient to restore BDNF-induced LTP (*Figure 8*). This is important because BDNF-induced LTP is a potent modulator of brain synaptic plasticity (*Lu et al., 2013*). Although alterations in BDNF signaling caused by disruptions in its expression or changes in levels of *Ntrk2* receptor isoforms have been associated with neurodegeneration, psychiatric disorders, intellectual disabilities and autism, most studies have focused on the mechanisms regulating *Bdnf* expression and not its receptor TrkB (*Qin et al., 2016*; *Zheng et al., 2016*). For example, it has been shown that at the transcription level *Bdnf* expression is regulated by epigenetic factors and via the use of multiple promoters. Different transcription factors can bind to these promoters to generate transcripts containing the unique BDNF protein encoding exon. In addition, the neuronal spatially and temporally regulated processing of the pro-BDNF peptide provides another level at which BDNF function can be modulated (Reviewed in *Hing et al. (2018)*). The significance of this second regulatory mechanism has been validated by the Val66Met *BDNF* polymorphism in humans which affects the processing and trafficking of BDNF and causes phenotypes including profound impairments in synaptic and cognitive functions (*Chen et al., 2005*; *Egan et al., 2003*; *Soliman et al., 2010*). However, almost nothing is known about the mechanisms regulating *Ntrk2* expression or the mechanisms leading to its pathological dysregulation (*Kemppainen et al., 2012*; *Wong et al., 2012*). Contrary to the *Bdnf* gene that has multiple promoters activated by different stimuli, the *Ntrk2* locus has only one promoter. The *NTRK2* gene spans over 350 Kb of DNA sequence and allows for the generation, by alternative splicing of at least 36 potential transcripts from 24 exons (*Luberg et al., 2010*). However, gene targeting experiments have shown that two isoforms, namely TrkB.FL and TrkB.T1, are the most highly expressed and phenotypically important during development [*Figure 4*; (*Dorsey et al., 2006*; *Fulgenzi et al., 2015*; *Klein et al., 1993*). To date, no RBPs have been shown to regulate *Ntrk2* splicing or mRNA

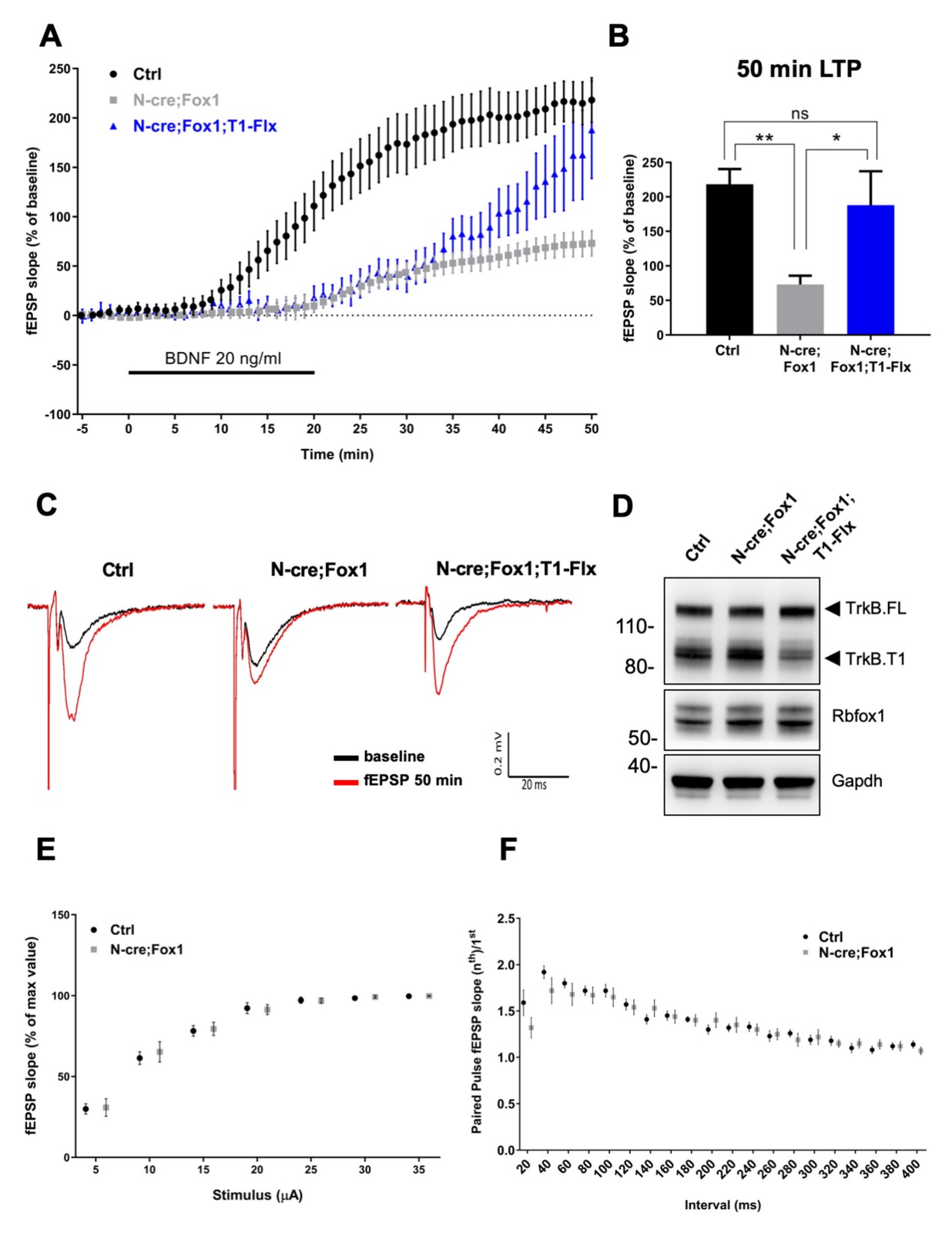

**Figure 8.** Rbfox1 overexpressing mice have impaired BDNF-induced LTP that is rescued by conditional removal of one TrkB. T1 receptor allele. (**A**) Averaged time-course of the field excitatory postsynaptic potential (fEPSP) slope in hippocampal slices from control (Ctrl = *Nes* Cre, n = 10), N-cre;Fox1 (*Nes-Cre;R26-Rbfox1*[+/flox], n = 10) and N-cre;Fox1;T1-Flx (*Nes*-Cre;*R26-Rbfox1*[+/flox]; TrkB.T1[+/flox], n = 8) mice after extracellular application of BDNF (20 ng/ml). Values are expressed as fEPSP percentage of the baseline values (average of 5 min before BDNF application, mean ± SEM). BDNF application
*Figure 8 continued on next page*

*Figure 8 continued*

(20 min) is indicated by the black horizontal bar. (B) Histogram showing mean ± SEM of the fEPSP slope average at 50 min after BDNF infusion; ** = p ≤ 0.01; * = p ≤ 0.05; ns = p > 0.05 (One-way ANOVA followed by Tukey's test). (C) Representative recording traces of slices before (baseline = black line) and 50 min after BDNF application (red line). (D) Western blot analysis of hippocampi from Ctrl, N-cre;Fox1 and N-cre;Fox1;T1-Flx mice probed with anti TrkB, Rbfox1 and Gapdh antibodies. (E) Input/Output relationship of Shaffer collateral projections on CA1 pyramidal neurons of Ctrl and N-cre;Fox1 hippocampal slices. Slope of fEPSP reported as percentage of the maximal recorded values plotted against the stimulation current. (F) Ratio between the second and first fEPSP evoked by the stimulation of Shaffer collateral and recorded in the CA1-radiatum at increasing time latency from 20 to 400 ms of Ctrl and N-cre;Fox1 hippocampal slices.

DOI: https://doi.org/10.7554/eLife.49673.015

The following figure supplements are available for figure 8:

**Figure supplement 1.** Rbfox1 overexpressing mice have impaired BDNF-induced LTP that is rescued by removal of one TrkB.

DOI: https://doi.org/10.7554/eLife.49673.016

**Figure supplement 2.** BDNF-induced LTP is not altered in TrkB.

DOI: https://doi.org/10.7554/eLife.49673.017

transcripts levels. Thus, *Rbfox1* is the first gene shown to regulate *Ntrk2* expression at the mRNA level. Its direct binding to *Ntrk2* RNA (*Figure 3C*) and its function in promoting TrkB.T1 RNA stability, rather than splicing (*Figure 3D*) are strongly supported by: 1) the in vivo and in vitro Rbfox1 overexpression leading to up-regulation of only TrkB.T1 and no changes in TrkB.FL; 2) the confirmation of the change in expression of only TrkB.T1 by the RNA-seq analysis in N-cre-Fox1 mouse hippocampus; 3) the analysis of new nascent RNA showing that TrkB.T1, but not TrkB.FL RNA is more stable upon *Rbfox1* overexpression. Nevertheless, the details of this newly identified molecular mechanism by which RbFox1 expression influences TrkB.T1 mRNA levels needs to be further elucidated since it appears to be independent of an action on the 3'UTR region (*Figure 3—figure supplement 1*). Importantly, this mechanism does not appear to be limited to TrkB.T1 since it has also been reported that in the hippocampus of *Rbfox1* knock-out mice some genes lacking iCLIP clusters or GCAUG binding elements in the 3'UTR region can still be differentially expressed (*Vuong et al., 2018*) and Rbfox1 can increase the mRNA concentration of genes that lack identified 3'UTR miRNA binding sites (*Lee et al., 2016*).

While upregulation of Rbfox1 changes TrkB.T1 expression, a reduction in its level does not have an effect. The upregulation of Rbfox2 supports a compensatory mechanism by this family member since both genes are expressed in neurons and like all Rbfox proteins they bind to the same (U) GCAUG motif. However, the fact that only overexpression of Rbfox2, but not Rbfox3, can upregulate TrkB.T1 in primary neurons (*Figure 2C,D*) suggest that different, yet unknown binding partners might be part of this mechanism. Mass spectrometry analysis of Rbfox protein complexes should help identifying which molecular players can differentiate Rbfox protein functions.

Upregulation of Rbfox1 has an inhibitory function on BDNF-induced synaptic plasticity most likely by reducing the TrkB kinase signaling through an increase of TrkB.T1 expression. This is in line with the observation that nervous system–specific deletion of *Rbfox1* results in heightened susceptibility to spontaneous and kainic acid–induced seizures which suggest increased neuronal excitability (*Gehman et al., 2011*). The presence of such regulatory mechanisms on TrkB function is important considering the increased evidence supporting a key role for TrkB activation in epileptogenesis caused by status epilepticus (*McNamara and Scharfman, 2012*). Thus, having TrkB activity under the control of *Rbfox1*, which also regulates the splicing of several synaptic function genes including ion channels, neurotransmitter receptors, structural proteins of the synapse and vesicle fusion proteins, suggests a unifying biological system to regulate neuronal function and excitation (*Gehman et al., 2011*; *McNamara and Scharfman, 2012*).

In humans, reduced *RBFOX1* expression has been associated with autism, epilepsy and heart disease although the mechanistic relationship between *RBFOX1* downregulation and these disorders is still unknown (*Bill et al., 2013*; *Gao et al., 2016*). Impaired BDNF signaling has also been associated with these diseases raising the interesting possibility that *RBFOX1* regulation of the expression of TrkB receptors might, at least in part, be part of the pathogenetic mechanism. Increasing or decreasing TrkB.T1-receptor levels indeed affects BDNF/TrkB signaling (*Figure 6*). Although, we do not see changes in TrkB.T1 expression in *Rbfox1* heterozygous mice, it is conceivable that the compensatory mechanisms between mice and humans are different. For example, *Nedd4-2* heterozygous mice

have minimal developmental abnormalities such as hyperactivity, increased basal synaptic transmission and enhanced sensitivity to inflammatory pain (*Yanpallewar et al., 2016*). However, human heterozygous missense mutations cause major malformations including periventricular nodular heterotopia leading to hypotonia, intellectual disability, seizures, syndactyly and cleft palate (*Broix et al., 2016*; *Kato et al., 2017*). In addition, *Ntrk2* heterozygous mice do not show major abnormalities while loss of one *NTRK2* (TrkB) allele in human leads to hyperphagic obesity and severe impairment in memory, learning and nociception (*Klein et al., 1993*; *Yeo et al., 2004*).

Most epidemiological studies associate gene loss of function caused by deletions or point mutations with disease. Indeed, even for *RBFOX1* most studies have looked at *RBFOX1* downregulation. However, our findings suggest the need to investigate causal relationship between *RBFOX1* gain of function to specific pathologies. For example, it has been reported that neurons derived from iPSCs of Parkinson's disease (PD) patients have elevated *RBFOX1* levels (*Lin et al., 2016*). Interestingly, there is already abundant data showing a role for BDNF/TrkB signaling in the etiology of Parkinson's disease, at least from in vivo animal models. For example, ablation of the *Bdnf* gene impairs the survival and/or maturation of substantia nigra (SN) dopamine (DA) neurons during development (*Baquet et al., 2005*); loss of one copy of *Ntrk2* leads to an age-dependent increase in the levels of α-synuclein in the SN (*von Bohlen und Halbach et al., 2005*), and in a mouse model with a chronic reduction in TrkB signaling (~30% of WT) there is an age-dependent and selective degeneration of SN DA neurons and increased vulnerability of these neurons to neurotoxins (*Baydyuk and Xu, 2014*). Because it has also been reported that neurons of PD patients have an increase in TrkB.T1 expression (*Fenner et al., 2014*), it will be of interest to test whether sporadic PD patients have *RBFOX1* upregulation and, if that is the case, an association with increased TrkB.T1 receptor levels. In another study, it has been reported that in the human population there is an intronic SNP affecting an enhancer in the fourth intron of *RBFOX1* that leads to a substantial increase in expression (*Carter et al., 2017*). While this SNP has been identified in the context of a study evaluating how inherited polymorphisms carried in the germline affect the somatic evolution of a tumor, it will be important to study whether this SNP has other effects on cognitive brain functions and leads to an increase in the incidence of neurodegenerative disorders where TrkB.T1 upregulation has also been reported (*Cai et al., 2006*; *Dwivedi et al., 2003*; *Ernst et al., 2009b*; *Karege et al., 2005a*; *Karege et al., 2005b*). Most importantly, the change in TrkB.T1 level caused by Rbfox1 upregulation is not dramatic, but it is occurring at the synaptic terminals (*Figure 7*) which in turn can compromise critical BDNF functions on synaptic plasticity (*Figure 8*). Examples where small changes in the expression of specific genes are very detrimental and relevant to human pathology have already been described. For instance, gain of function mutations that facilitate or enhance the activation of protein kinase Cγ (PKC) by only approximately 10% have been associated with Alzheimer's disease (*Alfonso et al., 2016*). PKC is required for the synaptic depression caused by amyloid-β (Aβ) and it has been suggested that a lifetime of slightly enhanced signaling may sensitize individuals to the detrimental effects of Aβ leading to AD (*Newton, 2018*).

In all, our study lays the foundation to investigate whether upregulation of *Rbfox1* leads to alteration of brain functions or neurodegenerative disorders that are ultimately associated with a dysregulation in BDNF/TrkB signaling, a pathway that has already been convincingly associated with normal development of the nervous system.

## Materials and methods

### Key resources table

| Reagent type | Description | Reference | Identifiers | Additional information |
|---|---|---|---|---|
| Genetic reagent (*M.musculus*) | *Rbfox1flox/flox* | (*Gehman et al., 2011*) Jackson Laboratory | IMSR Cat# JAX:014089, RRID:IMSR_JAX:014089 | |
| Genetic reagent (*M.musculus*) | *Nes*-cre | Jackson Laboratory | IMSR Cat# JAX:003771, RRID:IMSR_JAX:003771 | |
| Genetic reagent (*M.musculus*) | *Gt(ROSA)26Sor-LacZ* | Jackson Laboratory | IMSR Cat# JAX:003309, RRID:IMSR_JAX:003309 | |

*Continued on next page*

Continued

| Reagent type | Description | Reference | Identifiers | Additional information |
|---|---|---|---|---|
| Plasmid | CAG-STOP-eGFP-ROSA26TV | Addgene | RRID:Addgene_15912 | |
| Antibody | Anti-GFAP (rabbit polyclonal) | Agilent (Dako) | Agilent Cat# Z0334, RRID:AB_10013382 | IF (1:200) |
| Antibody | Anti-TrkB (rabbit polyclonal) | Millipore | Millipore Cat# 07–225, RRID:AB_310445 | WB (1:1000) |
| Antibody | Anti-TrkB (C13) (rabbit polyclonal) | Santa Cruz Biotechnology | Santa Cruz Biotechnology Cat# sc-119, RRID:AB_632559 | WB (1:500) |
| Antibody | Anti-TrkC (rabbit monoclonal) | Cell Signaling Technology | Cell Signaling Technology Cat# 3376, RRID:AB_2155283 | WB (1:1000) |
| Antibody | Anti-GAPDH (mouse monoclonal) | Millipore | Millipore Cat# MAB374, RRID:AB_2107445 | WB (1:2000) |
| Antibody | Anti-Cre (rabbit monoclonal) | Cell Signaling Technology | Cell Signaling Technology Cat# 15036, RRID:AB_2798694 | WB (1:1000) |
| Antibody | Anti-Rbfox1 (mouse monoclonal) | Millipore | Millipore Cat# MABE985, RRID:AB_2737389 | WB (1:1000); IF (1:200) |
| Antibody | Anti-Rbfox2 (rabbit polyclonal) | Bethyl Laboratories | Bethyl Cat# A300-864A, RRID:AB_609476 | WB (1:1000) |
| Antibody | Anti-Rbfox3 (mouse monoclonal) | Millipore | Millipore Cat# MAB377, RRID:AB_2298772 | WB (1:1000) |
| Antibody | Anti-Tra2b (rabbit polyclonal) | Bethyl Laboratories | Bethyl Cat# A305-011A, RRID:AB_2621205 | WB (1:1000) |
| Antibody | Anti-PSD95 (rabbit polyclonal) | Millipore | Millipore Cat# AB9708, RRID:AB_2092543 | WB (1:1000) |
| Antibody | Anti-phospho-TrkB (rabbit polyclonal) | Cell Signaling Technology | Cell Signaling Technology Cat# 9141, RRID:AB_2298805 | WB (1:1000) |
| Antibody | Anti-phospho-ERK (mouse monoclonal) | Cell Signaling Technology | Cell Signaling Technology Cat# 9106, RRID:AB_331768 | WB (1:1000) |
| Antibody | Anti-ERK (rabbit polyclonal) | Cell Signaling Technology | Cell Signaling Technology Cat# 9102, RRID:AB_330744 | WB (1:1000) |
| Antibody | Anti-β-Actin (mouse monoclonal) | Santa Cruz Biotechnology | Santa Cruz Biotechnology Cat# sc-47778 HRP, RRID:AB_2714189 | WB (1:3000) |
| Antibody | Anti-β-III-Tubulin (mouse monoclonal) | Covance | Covance Cat# MMS-435P, RRID:AB_2313773 | WB (1:1000) |
| Antibody | Donkey Anti-mouse Alexa Fluor 488 Conjugated | Thermo Fisher Scientific | Thermo Fisher Scientific Cat# A-21202, RRID:AB_141607 | IF (1:1000) |
| Antibody | Donkey Anti-rabbit Alexa Fluor 555 Conjugated | Thermo Fisher Scientific | Thermo Fisher Scientific Cat# A-31572, RRID:AB_162543 | IF (1:1000) |

## Mouse models

The *Rbfox1* transgenic mouse model (R26-Fox1) was generated by targeting the *Gt(ROSA)26Sor* locus (Chromosome 6) with a CTV vector (Addgene #15912) to conditionally express murine *Rbfox1*. *Rbfox1* cDNA (ID:6821627, Dharmacon) was cloned into the CTV vector using the AscI restriction

enzyme site. In the targeting vector, a removable STOP cassette flanked by loxP sites is present in between a CAG promoter and the *Rbfox1* cDNA (*Figure 1—figure supplement 2A*). The targeting vector was electroporated in the CJ7 embryonic stem cell line (129/sv), as previously described (*Tessarollo, 2001*), and recombinant clones were injected into C57BL/6J blastocysts to produce chimeras that transmitted the targeted *Gt(ROSA)26Sor* allele to the progeny. Animals were backcrossed into a pure C57BL/6J background for about 10 generations. TrkB.T1 conditional knockout mice (TrkB.T1flox/flox) are from *Dorsey et al. (2006)*, while *Rbfox1* conditional knockout (*Gehman et al., 2011*), *Nes*-cre (JAX strain 003771) and *Gt(ROSA)26Sor*-LacZ transgenic mice (JAX strain 003309) were obtained from the Jackson Laboratory. Animals were bred in a specific, pathogen-free facility with food and water *ad libitum*. All experimental procedures followed the National Institutes of Health Guidelines for animal care and use, and were approved by the NCI-Frederick Animal Care and Use Committee.

## X-Gal staining of brain sections (β-galactosidase activity)

*Gt(ROSA)26Sor*-LacZ mice and *Nes*-cre; *Gt(ROSA)26Sor*-LacZ mice were transcardially perfused with phosphate buffered saline (PBS), followed by PFA 4% in PBS at RT. Brains were dissected and cryoprotected in 30% sucrose solution (in PBS) overnight at 4°C. Floating brain sections (50 µm) were collected in PBS and stained with X-Gal staining solution for 5 min at 30°C in the dark and washed in PBS. X-Gal staining solution in PBS: 2 mg/ml X-Gal (ThermoFisher Scientific), 5 mM $K_3Fe(CN)_6$, 5 mM $K_4Fe(CN)_6$, 2 mM $MgCl_2$, 0.25% Triton X-100.

## Immunofluorescence of brain sections

Mice were perfused in PFA 4% in PBS and brains dissected and cryoprotected in 30% sucrose solution (in PBS) overnight at 4C. After sectioning, floating brain sections (50 µm) were collected in PBS, blocked 30 min at RT in blocking solution (10% normal donkey serum, 0.1% Triton X-100 in PBS) and incubated overnight at 4°C with primary antibodies. Sections were then washed three times with PBS and incubated with secondary antibodies in PBS for 2 hr at RT. Sections were washed again three times in PBS and stained with DAPI (Invitrogen) for 5 min at RT and imaged using Zeiss LCI 510 Meta confocal microscope. Primary antibodies were: anti-Rbfox1 (1D10, Millipore), anti-GFAP (Z0334, Dako). Secondary antibodies were: donkey-anti-mouse Alexa Fluor 488 (A21202, Invitrogen) and donkey-anti-rabbit Alexa Fluor 555 (A31572, Invitrogen).

## Hippocampal neuron cultures

Hippocampi used to generate primary neurons were collected from E17.5/E18.5 fetuses. After removal of the meninges from each cortex, hippocampi were dissected and collected in DMEM serum free media (Gibco). The hippocampal tissue was minced into small pieces, digested with Trypsin-EDTA (Gibco) at 0.125% for 25 min at 37°C, transferred into a new tube containing 2 ml of DMEM containing 10% FBS to inactivate trypsin and dissociated using a sterile Pasteur glass pipette flamed to slightly narrow the opening, triturated up and down 6–8 times. A last passage to facilitate single cell dissociation was done using a flame-pulled Pasteur glass pipette, triturating up and down 1–2 times. 80–100 µl of cell suspension was counted using Trypan Blue for live staining while the remaining cell suspension was centrifuged at 2000 rpm for 5 min. The cell pellet was re-suspended in DMEM with 10% FBS to obtain a concentration of $5 \times 10^5$ cells in 200 µl. $5 \times 10^5$ cells (200 µl of cell suspension) were pre-plated in the center of 35 mm dishes previously treated with Poly-D Lysine (P6407, Sigma Aldrich) and incubated for 2 hr at 37°C before adding 2 ml of Neurobasal media with B27 supplement (Gibco) to each dish. After 24 hr, AraC 1 µM (C6645, Sigma Aldrich) was added only once to the cultured neurons to avoid proliferation of dividing cells. Half of the Neurobasal media supplemented with B27 was replaced every 2 days.

Lentiviral expressing vectors GFP control, *Rbfox1* (ID:6821627, Dharmacon), *Rbfox2* (ID:93686, Dharmacon), *Rbfox3* (ID:52897, Dharmacon), *Tra2b* (ID:6594226, Dharmacon), together with adenoviral vectors expressing the recombinant Cre protein (Ad-Cre) were generated and obtained from the Protein Expression Laboratory (PEL) facility at the NCI-Frederick. Adenoviral vectors expressing Rbfox1 and mutant Rbfox1-F159A were obtained from Vigene Biosciences.

Primary neurons were starved in Neurobasal media without B27 supplement (Gibco) for 5 hr before BDNF treatment for 5 min (1 ng/ml; Alomone Labs, Cat# B-250).

## Western blot analysis

Mouse hippocampi from 2 to 3 months old mice were quickly dissected in cold PBS and lysed in Precellys ceramic lysing kit tube with 0.5 ml of RIPA lysis buffer (20–188, Millipore) using three cycles of 20 s/cycle at 5000 rpm in PRECELLYS 24 (Bertin Technologies). Lysates were then incubated 20 min at 4°C. After the incubation, lysates were centrifuged at top speed (13,000 rpm) using a table-top centrifuge at 4°C. Only the top cleared part of the lysates was collected and transferred into new tubes. The total amount of protein was then quantified using BCA assay (23225, ThermoFisher Scientific) and samples were prepared using the same amount of total protein before adding Laemmli sample buffer 2X (S3401, Sigma-Aldrich). The samples were heated at 95°C for 5 min before being loaded in 4–12% NuPAGE (ThermoFisher Scientific) precast gels for western analysis.

Primary hippocampal neurons were lysed by adding 150 µl of Laemmli sample buffer 2X directly into the culture dishes after being washed with cold PBS. The lysed neurons were then transferred into 1.5 ml tubes and sonicated using Bioruptor-300 (Diagenode) to shred genomic DNA and eliminate sample viscosity. Samples were then heated at 95°C for 5 min before being loaded in 4–12% NuPAGE precast gels for western blot analysis. After being transferred to PVDF membranes (LC2005, ThermoFisher Scientific), blots were blocked in 5% non-fat milk in TBS-Tween (0.1%) and incubated overnight at 4°C with specific antibodies. Antibodies were: anti-TrkB (against the extracellular domain of TrkB and therefore recognizing all TrkB isoforms; Millipore 07–225), anti-TrkB.T1 C13 (sc-119; Santa Cruz), anti-TrkC (C44H5, Cell Signaling), anti GAPDH (MAB374; Millipore), anti-Cre (Cre recombinase, D7L7L Cell Signaling), anti-Rbfox1 (1D10, a kind gift of Dr. Doug Black, and Millipore), anti-Rbfox2 (A300-864A Bethyl Laboratories), anti-Rbfox3 (MAB377, Millipore) anti-Tra2β (A305-011A Bethyl Laboratories) anti-PSD-95 (Millipore AB9708), anti-phospho-TrkB (#9141, Cell Signaling), anti-phospho-ERK (#9106, Cell Signaling), anti-ERK (#9102, Cell Signaling), β-Actin (Santa Cruz sc-47778) and anti β-III-Tubulin (Tuj1, Covance). After incubation with the appropriate horseradish peroxidase (HRP)-conjugated secondary antibodies (Millipore), membranes were incubated with enhanced chemilumescent substrate (34076, ThermoFisher Scientific) for detection of HRP enzyme activity and visualized in a Syngene gel documentation system. Bands in immunoblots were quantified by Syngene software. Student t-test was applied for statistical significance assessment.

## Synaptosomes preparation

Synaptosomes were obtained from the whole hippocampi of control animals (*Nes-Cre*; Ctrl) and N-cre;Fox1 animals (*Nes-Cre;R26-Rbfox1*$^{+/flox}$). Synaptosomes were isolated using Syn-PER Synaptic Protein Extraction Reagent (ThermoFisher Scientific Cat.No. 87793): each hippocampus was homogenized in 700 µl ice-cold Syn-PER reagent previously added with EDTA-free protease inhibitors (Roche Cat.No. 04 693 159 001) using a tissue homogenizer at low speed for 5 s. Homogenate samples (H) were collected (100 µl) before proceeding to centrifuge the remaining homogenate at 2.1 rcf/10 min/ +4°C. Supernantant (500 µl) was collected, transferred to a new tube and centrifuged again at 16.1 rcf/10 min/ +4°C. 400 µl of supernatant was collected as cytosolic fraction (C) while the pellet was re-suspended in 200 µl of ice cold Syn-PER reagent as synaptosome suspension (Syn). All the fractions (homogenates, cytosolic fractions, synaptosome suspensions) were then sonicated at +4°C using Bioruptor-300 (Diagenode) [30 s ON/30 s OFF for 10 cycles] before using BCA assay (23225, ThermoFisher Scientific) for total protein quantification. Samples were then prepared for western blot using equal amounts of total protein before adding Laemmli sample buffer 2X (S3401, Sigma-Aldrich) and loaded on 4–12% NuPAGE precast gels. Student t-test was applied for statistical significance assessment.

## qPCR

Total RNA was extracted from primary neurons or hippocampi using Qiagen RNeasy Mini kit (Cat.no 74104) according to manufacturer's instruction. cDNA was then generated using SuperScript III First-Strand Synthesis System (Cat. No 18080–051, ThermoFisher Scientific). Real time PCR was performed using BioRad iTaq Universal SYBR-green Supermix (Cat.No. 172–5120) in a MX3000P (Agilent Technologies) apparatus with the following program: 95°C for 3 min; 95°C 10 s, 60°C 20 s for 40 cycles; 95°C 1 min and down to 55°C (gradient of 1°C) for 41 cycles (melting curve step). Delta Ct values were obtained using GAPDH as reference gene.

Student t-test was applied for statistical significance assessment.

Primers used:
TrkB common forward: 5'-AGCAATCGGGAGCATCTCT-3'
TrkB.FL reverse: 5'-CTGGCAGAGTCATCGTCGT-3'
TrkB.T1 reverse: 5'-TACCCATCCAGTGGGATCTT-3'
GAPDH forward: 5'-TGCGACTTCAACAGCAACTC-3'
GAPDH reverse: 5'-ATGTAGGCCATGAGGTCCAC-3'
Rbfox1 forward: 5'- TGGCCCCAGTTCACTTGTAT-3'
Rbfox1 reverse: 5'- GCAGCCCTGAAGGTGTTGTA-3'

## RNA immunoprecipitation (RIP)

RNA immunoprecipitation was performed following the protocol from *Jayaseelan et al. (2011)*. Briefly, primary hippocampal neurons were cultured 4 days in vitro before being lysed using PLB buffer (100 mM KCl, 5 mM MgCl2, 10 mM HEPES pH 7, 0.5% Nonidet P-40, 1 mM DTT, 200 U/ml RNase OUT, 1 tablet of EDTA-free Complete Mini Protease Inhibitor). Protein-G magnetic beads (Dynabeads – ThermoFisher Scientific) were washed twice in NET-2 buffer (150 mM Tris-HCl pH 7, 100 mM Tris-HCl pH 8, 750 mM NaCl, 5 mM MgCl2, 0.25% NP-40, 20 mM EDTA pH 8, 1 mM DTT, 200 U/ml RNase OUT) and then conjugated with Rbfox1 (1D10) antibody overnight at 4°C. Beads/antibody slurry was washed six times using NT-2 buffer (150 mM Tris-HCl pH 7, 100 mM Tris-HCl pH 8, 750 mM NaCl, 5 mM MgCl2, 0.25% NP-40) and finally resuspended in 900 µl of NET-2 buffer for each sample. Primary neuron lysates were centrifugated at top speed (benchtop centrifuge) for 10 min at 4°C and 100 µl of cleared top lysate were added to each IP sample and incubated at 4°C overnight in rotation. Part of the initial samples (1:10) were collected as Inputs or Total samples. After the incubation, beads were washed six times with NT-2 buffer and then resuspended in 150 µl of Proteinase-K digestion buffer (NT-2 buffer supplemented with: 1% SDS, 1.2 mg/ml Proteinase-K) and incubated at 55°C for 30 min in a thermomixer.

RNA was then extracted by adding an equal volume (150 µl) of buffer saturated phenol-chloroform pH 4.5 with isoamyl alcohol, vortexed and centrifuged at 20.000 g for 10 min. The aqueous upper part was carefully collected in a new tube before adding 150 µl of chloroform, vortexed and centrifuged again. The upper part was again transferred in a new tube and 50 µl of 5 M ammonium acetate, 15 µl of 7.5 M LiCl, 5 µl of 5 mg/ml glycogen and 1 ml of ice cold 100% ethanol were added before placing the samples at −80°C overnight to allow RNA precipitation. Samples were then centrifuged at 20.000 g/30 min at 4°C, washed once with 80% ethanol and centrifuged again. RNA pellets were finally resuspended in 20 µl of RNase-free water.

RNA was polyadenylated using Poly(A) Tailing Kit (AM1350, ThermoFisher Scientific) following the manufacturer instruction.

cDNA was generated using SuperScript III First-Strand Synthesis System (Cat. No 18080–051, ThermoFisher Scientific).

RT-PCR was performed using BioRad iTaq Universal SYBR-green Supermix (Cat.No. 172–5120) in a MX3000P (Agilent Technologies) apparatus with the following program: 95°C for 3 min; 95°C 10 s, 60°C 20 s for 40 cycles; 95°C 1 min and down to 55°C (gradient of 1°C) for 41 cycles (melting curve step).

RT-PCR products were run in agarose 2% gel.
Primers used:
TrkB common forward: 5'-CGTGGTGGTGATTGCATCTG-3'
TrkB.FL reverse: 5'-CCATTGGAGATGTGGTGGA-3'
TrkB.T1 reverse: 5'-CAGTGGGATCTTATGAAACAAAACAA-3'
TrkB.T1-upstream intron forward: 5'-TTTGAGCATGACTTACGTTTCG-3'
TrkB.T1-upstream intron reverse: 5'-CCCAGCCTTTGTCTTTCCTT-3'
Camta1 forward: 5'-CCGGAGTTACAAGAAGTGTGG-3'
Camta1 reverse: 5'-CTTGGTCCTGCTTTTTGGTC-3'
Sirt1 forward: 5'-GAGCTGGATGATATGACGCTG-3'
Sirt1 reverse: 5'-CAGAGACGGCTGGAACTGTC-3'

## RNA stability

Primary hippocampal neurons for RNA stability studies were obtained by crossing *Nes*-Cre animals with *R26-Rbfox1*<sup>flox/flox</sup> animals. Embryos at E18.5 stage were dissected and fast genotyped by using EZ Fast Tissue Tail PCR Genotyping Kit (EZ Bioresearch) in order to group *Nes-Cre;R26-Rbfox1*<sup>flox/+</sup> hippocampi and R26-*Rbfox1*<sup>flox/+</sup> hippocampi. RNA stability was assessed in hippocampal neurons at 4 DIV using Click-iT Nascent RNA Capture Kit (C10365, ThermoFisher Scientific) following manufacturer's instruction. Briefly, 0.2 mM 5-ethynyl uridine (EU) ribonucleotide homolog was added to the neurons media to label new nascent RNA for 5 hr (pulse). After 5 hr, the media was replaced with normal media (without EU) for 9 hr (chase). Neurons were lysed at time 0 hr (right after 5 hr EU pulse) and at time 9 hr (after 9 hr of EU free media) and the RNA isolated using Qiagen RNeasy Mini kit (Cat.no 74104) according to manufacturer's instruction. Biotin azide was then chemically bound to the EU containing RNA molecules and streptavidin magnetic beads were used to capture the newly synthesized pool of RNA. cDNA was generated by using SuperScript VILO cDNA synthesis kit (11754–050, ThermoFisher Scientific).

Real time PCR was performed using BioRad iTaq Universal SYBR-green Supermix (Cat.No. 172–5120) in a MX3000P (Agilent Technologies) apparatus with the following program: 95℃ for 3 min; 95℃ 10 s, 60℃ 20 s for 40 cycles; 95℃ 1 min and down to 55℃ (gradient of 1℃) for 41 cycles (melting curve step). Delta Ct values were obtained using β-Actin as reference gene.

Student t-test was applied for statistical significance assessment.

Primers used:

TrkB common forward: 5'-AGCAATCGGGAGCATCTCT-3'

TrkB.FL reverse: 5'-CTGGCAGAGTCATCGTCGT-3'

TrkB.T1 reverse: 5'-TACCCATCCAGTGGGATCTT-3'

TrkC common forward: 5'-CCTGACACAGTGGTCATTGG-3'

TrkC.FL reverse: 5'-CTTGTCTTTGGTGGGGCTTA-3'

TrkC.T1 reverse: 5'-GACACATCCCCACTCTGGAC-3'

β-Actin forward: 5'-TACCACAGGCATTGTGATGG-3'

β-Actin reverse: 5'-TCTCAGCTGTGGTGGTGAAG-3'

## RNA seq analysis

RNA quality was assessed using an Agilent Bioanalyser. RNA integrity numbers (RIN) were observed to be greater than nine for all samples. Libraries were constructed from hippocampal total RNA using Illumina's TruSeq Stranded Total RNA Kit (RS-122–2201). Three biological replicates were used for each of the two experimental groups: Ctrl mice (*Nes-Cre*) and N-cre-Fox1 mice (*Nes-Cre;R26-Rbfox1*<sup>flox/+</sup>) of about 3 months of age and in the C57BL/6 background. Deep-sequencing was performed on an Illumina HiSeq 4000 in paired-end mode with a read length of 150 base-pairs.

The sequencing quality of the 99–153 million reads per sample was evaluated using FastQC (version 0.11.5), Preseq (version 2.0.3), Picard tools (version 1.119) and RSeQC (version 2.6.4). Reads were trimmed using Cutadapt (1.18) to remove adapter sequences, prior to mapping to the mm10 mouse reference genome using STAR (version 2.5.2b) in two-pass mode. Gene and transcript expression levels were quantified using RSEM (version 1.3.0) with gencode's M16 mouse annotation. EBSeq (version 1.22.1) was used to test for differential isoform expression between experimental conditions. Group-based TPM filtering was applied to remove lowly expressed transcripts. Significant differentially expressed isoforms were identified with a false-discovery rate ≤0.05. The R package clusterProfiler (version 3.10.1) was used for gene ontology (GO) enrichment analysis of the significant differential expressed gene-isoforms identified by EBSeq. Significant over-represented biological process GO terms were identified with a q-value less than 0.05. The RNA-seq data generated for this study have been deposited in NCBI's Gene Expression Omnibus and is accessible through GEO Series accession number GSE136253.

## iCLIP analysis

CLIP sequencing data (*Damianov et al., 2016*) (GSE76475) was downloaded from SRA using sratoolkit (version 2.9.2). Sequencing was done on Illumina HiSeq-2000 and pooled libraries were sequenced at a sequencing depth of ~15–18 million reads per sample. Sequencing quality was assessed using FastQC (version 0.11.5), Preseq (version 2.0.3), Picard tools (version 1.119), and

deeptools (version 2.5.0.1). Illumina sequencing adapters were trimmed from reads using cutadapt (version 1.14). Reads were aligned to the mouse genome version mm10 using BWA (version 0.7.15). Once the quality of the data was ensured, tracks suitable for viewing in IGV were downloaded from GEO (GSM1835189, GSM1835195). Bedops (version 2.4.30) and crossmap (version 0.2.7) were used to lift-over genomic coordinates from mm9 to mm10. BAM files were generated using bedtools (version 2.27.1).

## Cell lines

A *Rbfox1*-inducible cell line was established by using *Rbfox1* cDNA (ID:6821627, Dharmacon) and the Flp-In T-Rex 293 Cell Line system (R78007, ThermoFisher Scientific) according to the manufacturer instructions. *Rbfox1* expression was induced by Doxycycline (D3447, Millipore-Sigma) 0.5 μg/ml. TrkB.T1 cDNA including the complete 3'UTR sequence (plasmid pLTM665) was transfected into the *Rbfox1*-Flp-In T-Rex 293 cells using X-tremeGENE 9 DNA Transfection Reagent (6365779001, Millipore-Sigma).

Neuro-2a (N2A) neuroblastoma cells (ATCC, CCL-131) were transfected using TrkB.T1 cDNA including the complete 3'UTR sequence (plasmid pLTM665) in order to generate a stable N2A line expressing TrkB.T1 w/3'UTR. *Rbfox1* cDNA (ID:6821627, Dharmacon) was transiently transfected using X-tremeGENE 9 DNA Transfection Reagent (6365779001, Millipore-Sigma).

A HEK293 cell line with stable expression of TrkB.FL (HEK293-p618-2) was generated by transfecting HEK293 cells (ATCC, CRL-1573) with a TrkB.FL expression plasmid using X-tremeGENE 9 DNA Transfection Reagent (6365779001, Millipore-Sigma) and by puromycin selection. TrkB.FL expressing HEK293-p618-2 were transiently transfected with TrkB.T1 expressing plasmid by using X-tremeGENE 9 DNA Transfection Reagent (6365779001, Millipore-Sigma) and starved for 5 hr in serum free media (DMEM) before treatment with recombinant *BDNF*. All cell lines tested negative for mycoplasma.

## Electrophysiology

Coronal mouse brain slice for electrophysiological recording containing dorsal hippocampus was prepared according to *Ting et al. (2014)*. Briefly, 3 months-old mice were placed under deep Avertin anesthesia (250 mg/kg) and transcardially perfused with 25–30 mL of cold (10 C) carbogenated N-methyl-D-glucamine ACSF. Brains were extracted from the skull within 1 min, sectioned at 300 μm (Leica VT1200 vibratome) and appropriately incubated before recording (for all solutions and the complete procedure see section 2.1 on Materials and 3.1 Methods in *Ting et al., 2014*). For the electrophysiological recording, the slice was placed in a recording chamber under microscope (Zeiss Axioskop 2fs) and perfused with ACSF (2 ml/min; 28 C). A Teflon-coated concentric platinum–iridium electrode (FHC, ME USA) was placed in the stratum radiatum in the CA1 area of the dorsal HC, 300–400 μm from the recording electrode. Borosilicate glass recording electrodes were pulled (Sutter Instruments P90), ACSF filled to get 4–7 MΩ resistance, and placed in the apical dendritic region of CA1 pyramidal neurons. Field excitatory postsynaptic potentials (fEPSPs) were recorded in CA1 by activation of the Schaffer collaterals. An input-output curve was initially obtained by gradually increasing the stimulus intensity until the fEPSP reached a plateau. After which the stimulus was reduced to obtain an fEPSP that was 50% of the maximum level. Baseline recording was obtained by stimulating the slice every 20 s for approximately 45 min. Once the baseline was stabilized to obtain LTP, BDNF infusion was initiated and continued for 20 min (20 ng/ml; 2 ml/min). Recording was continued for 50 min. Field potential was recorded (Multiclamp 700b; Axon Instruments), digitized (10 kHz Digidata 1324), low-pass filtered (3 kHz, eight-pole Bessel), and stored (Clampex 9.2; Axon Instruments). Signals were analyzed off line (Clampfit 9.2; Axon Instruments), and the size of the fEPSP was evaluated by measuring the initial slope of the signal expressed as percentage of the variation from the baseline value (average of 5 min before the BDNF infusion). n = indicates the number of hippocampal slices analyzed (from ≥3 mice). One-way ANOVA followed by Tukey's multiple comparisons test was applied for statistical significance assessment.

## Statistics

Statistical significance was calculated using nonpaired two-tailed Student's t-test. One-way ANOVA followed by Tukey's multiple comparisons test was used for analysis of multiple groups (*Figure 8*,

*Figure 8—figure supplement 1*). All data are reported as mean ± SEM. ns = p > 0.05, * = p ≤ 0.05, ** = p ≤ 0.01, *** = p ≤ 0.001. n values for each individual experiment are indicated in the figure legends. GraphPad-Prism software was used to analyze data.

## Acknowledgements

We thank Eileen Southon for critical reading of the manuscript and Douglas Black for generously sharing the 1D10 anti-Rbfox1 monoclonal antibody. This work was supported by the NIH Intramural Research Program, Center for Cancer Research, National Cancer Institute.

## Additional information

### Funding

| Funder | Grant reference number | Author |
| --- | --- | --- |
| National Cancer Center | Intramural Research Program | Lino Tessarollo |

The funders had no role in study design, data collection and interpretation, or the decision to submit the work for publication.

### Author contributions

Francesco Tomassoni-Ardori, Conceptualization, Data curation, Investigation, Methodology, Writing—original draft, Writing—review and editing; Gianluca Fulgenzi, Conceptualization, Formal analysis, Investigation; Jodi Becker, Colleen Barrick, Mary Ellen Palko, Investigation, Performed experiments; Skyler Kuhn, Vishal Koparde, Formal analysis, Assisted with preparing manuscript; Maggie Cam, Resources, Supervision; Sudhirkumar Yanpallewar, Investigation, Writing—review and editing; Shalini Oberdoerffer, Formal analysis, Provided intellectual contribution; Lino Tessarollo, Conceptualization, Data curation, Supervision, Funding acquisition, Writing—original draft, Writing—review and editing

### Author ORCIDs

Lino Tessarollo (iD) https://orcid.org/0000-0001-6420-772X

### Ethics

Animal experimentation: All experimental procedures followed the National Institutes of Health Guidelines for animal care and use, and were approved by the NCI-Frederick Animal Care and Use Committee.

### Decision letter and Author response

Decision letter https://doi.org/10.7554/eLife.49673.026
Author response https://doi.org/10.7554/eLife.49673.027

## Additional files

### Supplementary files

- Supplementary file 1. List of gene-isoforms significantly changed by *Rbfox1* up-regulation in the hippocampus (Excel file; N-cre-Fox1).
DOI: https://doi.org/10.7554/eLife.49673.018

- Supplementary file 2. List of significantly dysregulated gene-isoforms that are overlapping between *Rbfox1* upregulation or knockout in the hippocampus (Excel tab: N-cre;Fox1 vs *Rbfox1*-KO) and complete list of gene-isoforms significantly changed by *Rbfox1* knockdown in the hippocampus (Excel tab: *Rbfox1*-KO; *Vuong et al., 2018*).
DOI: https://doi.org/10.7554/eLife.49673.019

- Transparent reporting form

DOI: https://doi.org/10.7554/eLife.49673.020

## Data availability

All data generated or analysed during this study are included in the manuscript and supporting files. Source data files have been provided for Table 1 and 2. The RNA-seq data generated for this study have been deposited in NCBI's Gene Expression Omnibus and is accessible through GEO Series accession number GSE136253.

The following dataset was generated:

| Author(s) | Year | Dataset title | Dataset URL | Database and Identifier |
|---|---|---|---|---|
| Tomassoni-Ardori F, Kuhn S, Koparde V, Tessarollo L | 2019 | Rbfox1 up-regulation impairs BDNF-dependent hippocampal LTP by dysregulating TrkB isoform expression levels | https://www.ncbi.nlm.nih.gov/geo/query/acc.cgi?acc=GSE136253 | NCBI Gene Expression Omnibus, GSE136253 |

The following previously published dataset was used:

| Author(s) | Year | Dataset title | Dataset URL | Database and Identifier |
|---|---|---|---|---|
| Damianov A, Lin C, Lee J, Martin KC, Black DL | 2016 | Co-regulation of splicing by Rbfox1 and hnRNP M | https://www.ncbi.nlm.nih.gov/geo/query/acc.cgi?acc=GSE71468 | NCBI Gene Expression Omnibus, GSE71468 |

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
