## [Decision Letter]

[Editors’ note: a previous version of this study was rejected after peer review, but the authors submitted for reconsideration. The first decision letter after peer review is shown below.]

Thank you for submitting your work entitled "RbFox1 up-regulation impairs BDNF-dependent hippocampal LTP by dysregulating TrkB isoform expression levels" for consideration by *eLife*. Your article has been reviewed by three peer reviewers, and the evaluation has been overseen by a Reviewing Editor and a Senior Editor. The following individual involved in review of your submission has agreed to reveal their identity: Clive Bramham (Reviewer #1). Our decision has been reached after consultation between the reviewers. Based on these discussions and the individual reviews below, we regret to inform you that your work will not be considered further for publication in *eLife*.

Summary:

This study focuses upon the regulation of a truncated TrkB species, TrkB.T1, by the RNA binding protein RbFox1. There was agreement among the three reviewers that the regulation of TrkB.T1 by RbFox1 is of considerable interest. However, a number of reservations were raised. The major issues are the need for more evidence for the role of truncated TrkB in BDNF-dependent LTP; a resolution of how the levels of RbFox1 impact upon TrkB.T1 expression; and whether there is a non-neuronal contribution of Rbfox1. These are valid concerns that will require more experimentation in order to validate the major conclusions. Specific comments by the reviewers follow this letter. Based upon the common concerns and recommendations from the referees, we have therefore decided to decline the manuscript.

Reviewer #1:

The study provides important insights into the mechanism and functional regulation TrkB signaling via selective modulation in the expression of truncated TrkB.T1 receptors at synaptic sites. Regulation of TrkB.T1 is mediated by the RNA-binding protein RbFox1. In a gain-of-function model, RbFox1 stabilizes TrkB.T1 mRNA and increases expression of TrkB.T1 protein. Surprisingly, stabilization occurs through an unknown mechanism involving binding of RbFox1 to the intronic region of TrkB.T1 pre-mRNA, and not by binding to cis-acting elements in the 3'UTR. Functional studies in acute hippocampal slices show that transgenic overexpression of TrkB.T1 greatly inhibits LTP in the induced by rapid bath perfusion of BDNF. BDNF-LTP is partly restored in heterozygous transgenic mice, strongly supporting a role of TrkB.B1 in regulating TrkB function and plasticity in the adult hippocampus. The background for this work is the observation of enhanced TrkB.T1 expression in chromosome 16 trisomy mice. By identifying regulation of TrkB.T1 by RbFox1 and showing differences between gain and loss-of-function of the RNA binding-protein, the study opens new avenues for pre-clinical translational research.

1) The effect on BDNF- LTP requires substantiation. Transgenic expression of TrkB.T1 modulates the response. To better evaluate the physiological role of TrkB.T1, BDNF-LTP should also be assessed in TrkB.T1 knockouts. TrkB inhibitors should be applied to confirm a TrkB-mediated response.

2) Data on the binding sites and mechanism of RNA stabilization would improve the paper.

Reviewer #2:

This manuscript presents interesting data examining the consequences of RbFox1 upregulation as an approach to identify downstream genes regulated by RbFox1 which may have significance for brain disorders associated with RbFox1 mutation. The authors begin by following up on their prior Trisomy 16 work, to identify RbFox1 as a chromosome 16-encoded gene that could produce their previously observed upregulated expression of a truncated form of the TrkB receptor, TrkB.T1. Data is presented to show that viral overexpression of RbFox1 in hippocampal neurons produces upregulation of TrkB.T1 protein and mRNA without changing levels of full-length TrkB (TrkB.FL), and differential binding of RbFox1 to TrkB mRNAs is assessed. A mouse model with inducible RbFox1 expression is used to assess effects on gene expression by immunoblotting and RNA-seq and to probe BDNF-induced LTP. The authors compare their RbFox1 induced expression RNA-seq dataset with a published dataset from another group studying the effects of RbFox1 deletion (Vuong et al., 2018) to support a main conclusion that genes affected by RbFox1 gain of function differ from genes affected by RbFox1 loss of function.

Several aspects of the data make it difficult to definitively conclude that gain and loss of RbFox1 produce regulation of different genes:

1) The comparisons of the results of RbFox1 overexpression and RbFox1KO in Figure 4B are hampered by the fact that there may be many differences in experimental design, mouse genetic background, sample preparation, sequencing, analysis and filters applied, etc., between the authors' dataset and data presented in Vuong et al., 2018. It's not clear that differences in seq data will all or even mostly be due to biological differences in the effects of Rbfox1 overexpression compared to Rbfox1 deletion. Insufficient details are provided in this manuscript to compare even major biological variables such as the age of mice used by the authors for the RNA-seq experiments.

2) A proposed mechanism for the difference in genes regulated by gain or loss of RbFox1 expression is incompletely defined. The authors show that an RbFox1 heterozygote with reduced RbFox1 expression has an increase in RbFox2 expression relative to a control mouse, while RbFox1 gain of function does not alter RbFox2 levels. It is not clear whether RbFox2 is changed in the RbFox1 deletion context (Vuong et al., 2018) though, or to what extent RbFox2 can interchange for RbFox1 binding sites and gene regulation.

If RbFox1 differentially elevates levels of newly synthesized TrkB.T1 mRNA compared to TrkB.FL mRNA (Figure 2D), then can the authors explain why this action of RbFox1 overexpression fails to produce any change in the level of TrkB.T1 protein (Figure 3—figure supplement 1) in the HEK293 dox-inducible cell line?

Greater explanation is needed for the determination of reads as TrkB.T1 compared to TrkB.FL, especially given that TrkB.FL contains the sequence of TrkB.T1.

In the BDNF-induced LTP experiments, it is not clear that the effect of 'inactivation' of one TrkB.T1 allele is specific to the context of RbFox1 upregulation. This manipulation should be symmetrically applied in the context of to the Ctrl mouse as well. The data presented are insufficient to conclude that RbFox1 upregulation leads to a deficit in the BDNF-induced LTP resulting from an increase in TrkB.T1 expression (as opposed to indicating possible distinct roles for both TrkB.T1 and RbFox1 upregulation in BDNF-induced LTP).

Reviewer #3:

In this manuscript, Tomassoni-Ardori et al. investigate the impact of Rbfox1 on the expression of different TrkB isoforms. The findings are somewhat interesting though the overall impact is somewhat limited by the many other recent manuscripts studying the function of this protein and the focus on Rbfox1 overexpression even though disease is associated with reduced functional Rbfox.

However, the work does provide new insight into the potential role of Rbfox1 and the data demonstrating it binds and regulates TrkB.T1 is pretty conclusive, though there are a few shortcomings that should be addressed that are outlined below.

1) In the second paragraph of the Results, the authors state that the observation that Rbfox1 specifically impacts TrkB.T1 mRNA without affecting full length suggests that it does not act by regulating splicing. I do not really understand this argument and think conducting experiments to rule out an effect on splicing (or some additional explanation) are warranted.

2) It would be helpful to confirm that the upregulation of Rbfox1 in brains of lox-stop-lox/nestin-Cre animals is limited to neurons. Depending on what nestin-cre driver was used, it is possible that Rbfox1 is also upregulated in non-neuronal cells which could contribute to the gene expression changes observed.

3) In order to demonstrate that TrkB.T1 was a key target of Rbfox1 despite the many other targets was to use a BDNF induced LTP paradigm. As responses in this setup would be expected to be dominated by BDNF/TrkB signaling, any regulator of TRkB levels would be expected to have strong phenotype here and dominate over the impact of other genes. Perhaps another complimentary functional experiment, if possible would be helpful here.

4) Related to the above, it could be interesting to determine whether any of the other RNA-seq hits were TrekB dependent, this could be another opportunity to support a key role for Rbfox1 dependent regulation of TrkB.T1 in downstream phenotypes.

5) The authors comment on potential differences between mouse and human, and it might be helpful to confirm that Rbfox1 similarly regulates TrkB.T1 in human neurons and determine whether knockdown of Rbfox1 is effective in that system.

[Editors’ note: what now follows is the decision letter after the authors submitted for further consideration.]

We are pleased to inform you that your article, "RbFox1 up-regulation impairs BDNF-dependent hippocampal LTP by dysregulating TrkB isoform expression levels", has been accepted for publication in *eLife*. A summary of the study and the reviewers' comments are included below.

The BDNF TrkB receptor is responsible for synaptic plasticity, obesity and learning and memory. In addition to the full length TrkB tyrosine kinase receptor, a truncated form of TrkB (TrkB.T1) exists in the central nervous system. However the regulation of TrkB.T1 receptors is not well understood. In this study, the RNA binding protein, RBFox1, was found to upregulate the expression of TrkB.T1, which resulted in a deficit in BDNF-dependent LTP. This paper provides a new mechanism to explain the effects of BDNF upon synaptic plasticity.

The revised manuscript has been evaluated by two prior reviewers. Both reviewers found the revised manuscript adequately addressed the previous criticisms and questions with additional experiments and analysis. They concluded the revision is suitable for publication. One remaining question was whether the regulation of truncated TrkB occurred through RNA stability or alternatively by a splicing event. this issue can be dealt with in the future, but it would be useful to acknowledge it in the text of the paper.

Reviewer #1:

In this manuscript, Tessarollo and colleagues identify that RbFox1 regulates levels of TrkB.T1, a TrkB isoform that regulates activity of TrkB.FL. The then go on to show that upregulation RbFox1 impacts LTP specifically through regulating TrkB/T1 levels.

I was a reviewer of the last version of this manuscript, and the changes made by the authors have adequately addressed many of the concerns raised with the original version. The revised manuscript represents a more cohesive story with a logical flow that was harder to see in the previous version.

There are still a few issues with the manuscript that should be addressed that are outlined below, but in general the revised manuscript is greatly improved and should be considered for publication.

1) Despite additional experiments conducted by the authors, the mechanism by which RbFox1 specifically regulates TrkB.T1 is still a mystery. In the eyes of this reviewer, the data does not necessarily confirm an RNA stability mechanism and could be due to regulation of splicing to favor the T1 UTR. Altered splicing to include the T1 UTR could result in more T1 mRNA after RbFox overexpression in the pulse chase experiment (Figure 3D) and is consistent with the observation that the 3'UTR does not increase mRNA stability (Figure 3—figure supplement 2). Additional experiments to look at splicing directly and/or experiments looking at stability using full mRNAs for TrkB.T1 and TrkB.FL would be ideal, but depending on the views of other reviewers it could also be acceptable to note that in the text that either mechanism is possible.

2) The authors have done an excellent job demonstrating that (A) the LTP deficits following RbFox1 upregulation are Trkb.T1 dependent and (B) that RbFox2 is at least semi-redundant with RbFox1. Therefore, the gene expression data shown by the authors is not likely to underlie any of the core phenotypes of interest in the study. Because of this, it may be best to put less emphasis on this data through removing/rewording the significance statement for example.

Reviewer #2:

The authors have done a thorough revision of their manuscript, which is much improved from the prior submission. I believe that the manuscript is suitable for *eLife*.

While the mechanism of action by which RbFox1 overexpression upregulates truncated TrkB.T1 remains unclear, the authors now show that this action is also achieved by RbFox2 but not RbFox3.

It should be made clear within the manuscript text and figure legend whether the experiments in Figure 3—figure supplement 1 utilize a construct containing only the TrkB.T1 3'UTR, or the entire cDNA transcript inclusive of the 3'UTR.

The authors have made gains in further understanding the relationship of RbFox1, TrkB.T1, and BDNF-dependent LTP. They have added data that TrkB.T1 deficient cultured hippocampal neurons display an increased response to BDNF. The manuscript also now includes data showing that conditional removal of one TrkB.T1 receptor allele in animals with up-regulated RbFox1 is sufficient to rescue hippocampal BDNF dependent LTP, suggesting that TrkB is a relevant target of RbFox1 in BDNF-dependent LTP.

This would be a more appropriate conclusion than the more broad: 'genetic reduction of the TrkB.T1 isoform in animals with up-regulated RbFox1 is sufficient to restore hippocampal BDNF dependent LTP, suggesting that TrkB is a major target of RbFox1 pathological dysregulation' particularly since the author's observe a broad variety of gene dysregulation in their RNA-seq experiments.

Statistical analysis and description of experimental design are improved. Analysis of RNA-seq data is improved for consistency across datasets being compared, although it remains unclear what biological variables exist between the authors' dataset and the previously published comparison dataset.

---

## [Author Response]

[Editors’ note: the author responses to the first round of peer review follow.]

Summary:This study focuses upon the regulation of a truncated TrkB species, TrkB.T1, by the RNA binding protein RbFox1. There was agreement among the three reviewers that the regulation of TrkB.T1 by RbFox1 is of considerable interest. However, a number of reservations were raised. The major issues are the need for more evidence for the role of truncated TrkB in BDNF-dependent LTP; a resolution of how the levels of RbFox1 impact upon TrkB.T1 expression; and whether there is a non-neuronal contribution of Rbfox1. These are valid concerns that will require more experimentation in order to validate the major conclusions. Specific comments by the reviewers follow this letter. Based upon the common concerns and recommendations from the referees, we have therefore decided to decline the manuscript.

Thank you very much for the constructive comments that were provided by the whole reviewing team to our previous submission. The comments are all valid and constructive and we felt the need to thoroughly address them through the use of new mouse models and experiments to strengthen and improve our manuscript. Specifically:

1) We have generated new information on the role of TrkB.T1 in BDNF-induced LTP by providing mechanistic information showing that up or down-regulation of TrkB.T1 modulates TrkB.FL signaling (new Figure 6), which has been shown to be the primary determinant of BDNF-induced LTP. In addition, we have performed BDNF-induced LTP experiments in a new triple transgenic mouse model (nestin- cre; R26-Rbfox1; TrkB.T1^loxP^) to demonstrate a direct role of TrkB.T1 function in neurons (new Figure 8), where RbFox1 is also expressed (new Figure 8—figure supplement 2).

2) To further understand how the levels of RbFox1 impact TrkB.T1 expression we have investigated the specificity of this function within the RbFox1 family members. The surprising finding that only RbFox1 and RbFox2, but not RbFox3, regulate TrkB.T1 despite the fact that all Fox proteins bind to the same RNA sequence suggests that other proteins are required for this phenomena (new Figure 2C, D). This observation is important because it steers the RNA biology field toward the investigation of a new molecular mechanism of RbFox protein function. In addition we have further strengthened our finding that RbFox1 gain or loss of function affects different genetic pools by re-doing our bioinformatics analysis starting from raw data from all studies and by employing the same parameters and filtering (new Figure 5).

3) We have validated that RbFox1 function is strictly neuronal by performing a detailed characterization of hippocampal nestin-cre/R26-RbFox1 activity (new Figure 1—figure supplement 2) and by performing BDNF- induced LTP experiments in a new transgenic nestin-cre; R26- Rbfox1; TrkB.T1^loxP^ mouse model to demonstrate a direct role of TrkB.T1 function in neurons (new Figure 8).

Reviewer #1:The study provides important insights into the mechanism and functional regulation TrkB signaling via selective modulation in the expression of truncated TrkB.T1 receptors at synaptic sites. Regulation of TrkB.T1 is mediated by the RNA-binding protein RbFox1. In a gain-of-function model, RbFox1 stabilizes TrkB.T1 mRNA and increases expression of TrkB.T1 protein. Surprisingly, stabilization occurs through an unknown mechanisms involving binding of RbFox1 to the intronic region of TrkB.T1 pre-mRNA, and not by binding to cis-acting elements in the 3'UTR. Functional studies in acute hippocampal slices show that transgenic overexpression of TrkB.T1 greatly inhibits LTP in the induced by rapid bath perfusion of BDNF. BDNF-LTP is partly restored in heterozygous transgenic mice, strongly supporting a role of TrkB.B1 in regulating TrkB function and plasticity in the adult hippocampus. The background for this work is the observation of enhanced TrkB.T1 expression in chromosome 16 trisomy mice. By identifying regulation of TrkB.T1 by RbFox1 and showing differences between gain and loss-of-function of the RNA binding-protein, the study opens new avenues for pre-clinical translational research.1) The effect on BDNF- LTP requires substantiation. Transgenic expression of TrkB.T1 modulates the response. To better evaluate the physiological role of TrkB.T1, BDNF-LTP should also be assessed in TrkB.T1 knockouts. TrkB inhibitors should be applied to confirm a TrkB-mediated response.

We do agree with the reviewer that more experiments to strengthen the role of TrkB.T1 modulation in LTP ware important. Experiments with Trk inhibitors have already been performed and have suggested a key role for TrkB.FL signaling in BDNF induced LTP (Kang and Schuman, 1995). In addition, in our initial paper (Carim-Todd et al., 2009) where we evaluated LTP in TrkB.T1 KO mice we failed to detect any significant difference. Here, we have also reported that TrkB.T1 heterozygous mice do not show any defect in BDNF induced LTP (Figure 8—figure supplement 2). Since a key point of the paper is to evaluate whether neuronal TrkB.T1 upregulation (since RbFox1 is expressed only in neurons), disrupts BDNF-induced LTP we did test whether it is the TrkB.T1 up-regulation in neurons causing the impairment (a point also raised by reviewer 2 and 3). Therefore, we have generated a triple transgenic mouse with a conditional TrkB.T1^loxP^, R-26-RbFox1 and nestin-cre allele to test rescue of the BDNF induced LTP in neurons. The strategy led to a rescue (new Figure 8). Furthermore, to mechanistically address the role of TrkB.T1 modulation on TrkB.FL function, we have used an in vitro system and showed that increased TrkB.T1 levels reduced TrkB.FL and ERK phosphorylation levels (new Figure 6A). A modulatory role was also confirmed in pure TrkB.T1 KO primary hippocampal cultures where TrkB.FL signaling was increased (new Figure 6B-D).

2) Data on the binding sites and mechanism of RNA stabilization would improve the paper.

This is a valid point. So far, the data that we have accumulated show

1) by analysis of the iCLIP data, that RbFox1 associates along the TrkB transcripts in two different nuclear fractions of the mouse brain (Figure 3A). Direct binding is also confirmed by RIP analysis in primary hippocampal neurons (Figure 3C); 2) mechanistically, RbFox1 stabilizes specifically TrkB.T1 transcripts (Figure 3D) and 3) RbFox1 RNA-binding capability is required for regulation of TrkB.T1 (Figure 3E, F). (we have clarified this concept in the third paragraph of the Discussion). Importantly, during the revision process, we have found that in primary hippocampal neurons RbFox2, but not RbFox3 can substitute for RbFox1 (new Figure 2C, D). The implications of this finding are important because they suggest that other proteins binding to RbFox1 and/or RbFox2 confer specificity of action since all RbFox members bind to the same sequence ((U)GCAUG). We have added this important point in the Discussion (fourth paragraph). We think that future IP and proteomic analysis to identify proteins associated with the different RbFox gene products should help determine the identity of these proteins.

Reviewer #2:This manuscript presents interesting data examining the consequences of RbFox1 upregulation as an approach to identify downstream genes regulated by RbFox1 which may have significance for brain disorders associated with RbFox1 mutation. The authors begin by following up on their prior Trisomy 16 work, to identify RbFox1 as a chromosome 16-encoded gene that could produce their previously observed upregulated expression of a truncated form of the TrkB receptor, TrkB.T1. Data is presented to show that viral overexpression of RbFox1 in hippocampal neurons produces upregulation of TrkB.T1 protein and mRNA without changing levels of full-length TrkB (TrkB.FL), and differential binding of RbFox1 to TrkB mRNAs is assessed. A mouse model with inducible RbFox1 expression is used to assess effects on gene expression by immunoblotting and RNA-seq and to probe BDNF-induced LTP. The authors compare their RbFox1 induced expression RNA-seq dataset with a published dataset from another group studying the effects of RbFox1 deletion (Vuong et al., 2018) to support a main conclusion that genes affected by RbFox1 gain of function differ from genes affected by RbFox1 loss of function.Several aspects of the data make it difficult to definitively conclude that gain and loss of RbFox1 produce regulation of different genes:1) The comparisons of the results of RbFox1 overexpression and RbFox1KO in Figure 4B are hampered by the fact that there may be many differences in experimental design, mouse genetic background, sample preparation, sequencing, analysis and filters applied, etc., between the authors' dataset and data presented in Vuong et al., 2018. It's not clear that differences in seq data will all or even mostly be due to biological differences in the effects of Rbfox1 overexpression compared to Rbfox1 deletion. Insufficient details are provided in this manuscript to compare even major biological variables such as the age of mice used by the authors for the RNA-seq experiments.

The reviewer raises a very valid point. Therefore, we have completely redone the comparative analysis of RNA-seq data using raw data from all studies, using the same parameters, and by applying the same filtering during the analysis. In addition, we have analyzed the changes in isoform expression and used the same cut-off between studies. This analysis greatly increased the number of genetic changes caused by RbFox1 loss of function (1657 from 967 of the previous submission, new Figure 5B) yet only 42 (from 38 of previous submission) overlap between that genetic model and our gain-of-function study. We have also changed our analysis to report changes in pathways rather than individual genes (new Figure 5A) which we think is more relevant to understanding the implications on brain function.

We apologize for not having included all the details about the mice used for the analysis. We have now amended the Materials and methods section (subsection “Mouse models”; subsection “RNA stability”) to indicate that mice were backcrossed into a pure C57BL/6 genetic background for at least 10 generations and the nestin-cre mice were from the same source (JAX) as reported in the study by Vuong et al., 2018. The age of the animals is also indicated. Lastly, we have added more details in the Materials and methods section for the RNA-seq analysis (subsection “RNA seq analysis”). While we cannot completely rule out differences between studies we feel that, if any, these should be minimal after the new analysis.

2) A proposed mechanism for the difference in genes regulated by gain or loss of RbFox1 expression is incompletely defined. The authors show that an RbFox1 heterozygote with reduced RbFox1 expression has an increase in RbFox2 expression relative to a control mouse, while RbFox1 gain of function does not alter RbFox2 levels. It is not clear whether RbFox2 is changed in the RbFox1 deletion context (Vuong et al., 2018) though, or to what extent RbFox2 can interchange for RbFox1 binding sites and gene regulation.

As indicated in our response to point 2 of reviewer 1 we have now shown that RbFox2 is not only upregulated in RbFox1 heterozygous mice but can also functionally compensate for RbFox1 (Results, fourth paragraph; new Figure 2C, D; Discussion, fourth paragraph). As suggested by the reviewer, we have also checked whether RbFox2 is up-regulated in the context of RbFox1 deletion (Vuong et al., 2018) and found that indeed it is. We have added the reference (Results, fifth paragraph), and also reported the data in Supplementary file 2. In addition, RbFox1 and RbFox3 knockdown experiments in primary neurons also show a 2- fold increase in RbFox2 protein (Lee et al., 2016), as also previously reported in RbFox1-KO brain (Gehman et al., 2011).

Indeed, as also indicated in our response to reviewer 1, point 2, we have made some progress with the new data toward the understanding of what might regulate the stability of different transcripts in a overexpression context but we think that showing the biological significance and importance of this new mechanism should be the first critical step to prompt the RNA biology community to dissect the molecular basis of this important phenotype.

If RbFox1 differentially elevates levels of newly synthesized TrkB.T1 mRNA compared to TrkB.FL mRNA (Figure 2D), then can the authors explain why this action of RbFox1 overexpression fails to produce any change in the level of TrkB.T1 protein (Figure 3—figure supplement 1) in the HEK293 dox-inducible cell line?

We apologize for not having been more clear in explaining the HEK293-dox inducible experiment and its significance. Since it has been reported that RbFox1 can stabilize mRNA species by binding to the 3’UTRs and antagonize specific miRNA activity (Lee et al., 2016), we have tested if RbFox1 could increase the stability of TrkB.T1 cDNA with the 3’UTR. Our data show that RbFox1 presence does not change the level of TrkB.T1 with the 3’UTR suggesting that it is the binding to the intronic regions of the TrkB.T1 that changes its level of expression. This is in agreement with our iCLIP analysis and RIP experiments that show Rbfox1 binding to the intronic regions of TrkB.T1 (Figure 3A, C). We have now clarified the concept in the manuscript (Results and Discussion, third paragraph).

Greater explanation is needed for the determination of reads as TrkB.T1 compared to TrkB.FL, especially given that TrkB.FL contains the sequence of TrkB.T1.

The reviewer is correct that more details are needed for how we determined the reads of TrkB.T1 and TrkB.FL since there is sequence overlap between the two isoforms. Even if TrkB.FL and TrkB.T1 share the same extracellular region, the unique TrkB.T1 exon and long 3’UTR provides specificity for the determination of reads as TrkB.T1 compared to TrkB.FL. We have added more details in the Materials and methods section about the RNA-seq analysis including the use of “EBSeq (version 1.22.1) to test for differential isoform expression between experimental conditions. Group-based TPM filtering was applied to remove lowly expressed transcripts. Significant differentially expressed isoforms were identified with a false-discovery rate £0.05”.

In the BDNF-induced LTP experiments, it is not clear that the effect of 'inactivation' of one TrkB.T1 allele is specific to the context of RbFox1 upregulation. This manipulation should be symmetrically applied in the context of to the Ctrl mouse as well. The data presented are insufficient to conclude that RbFox1 upregulation leads to a deficit in the BDNF-induced LTP resulting from an increase in TrkB.T1 expression (as opposed to indicating possible distinct roles for both TrkB.T1 and RbFox1 upregulation in BDNF-induced LTP).

We thank the reviewer for raising this very important point that also helped in addressing point 1 of reviewer 1. Specifically, we have generated a triple transgenic mouse with a conditional TrkB.T1^loxP^, R-26-RbFox1 and nestin-cre alleles to test the rescue of the BDNF induced LTP in the same cellular context where RbFox1 is upregulated (neurons); see new Figure 8. All manipulations are now symmetrically applied. The N-cre-Fox1 mouse had the R-26-RbFox1 and nestin-cre alleles, and the control (Ctrl) had the nestin-cre only allele. As indicated above, we also went further to better dissect the role of the TrkB.T1-isoform in the context of BDNF/TrkB signaling by showing that increasing TrkB.T1 levels reduces BDNF- induced signaling (new Figure 6).

Reviewer #3:In this manuscript, Tomassoni-Ardori et al. investigate the impact of Rbfox1 on the expression of different TrkB isoforms. The findings are somewhat interesting though the overall impact is somewhat limited by the many other recent manuscripts studying the function of this protein and the focus on Rbfox1 overexpression even though disease is associated with reduced functional Rbfox.

The reviewer is correct that most studies have focused on RbFox1 loss of function though it is now emerging that there are pathological situations in humans where RbFox1 (and even RbFox2) is found to be up-regulated (see also comment above). Thus, we feel that we are covering a new understudied and yet important area of research.

However, the work does provide new insight into the potential role of Rbfox1 and the data demonstrating it binds and regulates TrkB.T1 is pretty conclusive, though there are a few shortcomings that should be addressed that are outlined below.1) In the second paragraph of the Results, the authors state that the observation that Rbfox1 specifically impacts TrkB.T1 mRNA without affecting full length suggests that it does not act by regulating splicing. I do not really understand this argument and think conducting experiments to rule out an effect on splicing (or some additional explanation) are warranted.

We do understand the difficulty of this concept especially because RbFox1 is known as a regulator of splicing. The data that lead to our conclusion are the following: 1) in vivo and in vitro RbFox1 overexpression causes up-regulation of only TrkB.T1 and no changes in TrkB.FL. In situations of alternative splicing we should have seen TrkB.FL expression levels going down; 2) These data, i.e. change of only one isoform, is confirmed by the very sensitive RNA-seq analysis in N-cre-Fox1 mouse hippocampus. 3) Analysis of new nascent RNA showed that TrkB.T1 but not TrkB.FL RNA is more stable upon RbFox1 overexpression. We have now modified the Discussion to clarify these points (third paragraph).

2) It would be helpful to confirm that the upregulation of Rbfox1 in brains of lox-stop-lox/nestin-Cre animals is limited to neurons. Depending on what nestin-cre driver was used, it is possible that Rbfox1 is also upregulated in non-neuronal cells which could contribute to the gene expression changes observed.

We thank the reviewer for bringing up this very important point. As suggested, we have performed a very careful analysis of nestin-cre activity in the hippocampus. The characterization of cre-activity as well as of Fox1 expression and potential overlap with GFAP is reported in the new panels of Figure 1—figure supplement 2 and in the Results section (third paragraph).

3) In order to demonstrate that TrkB.T1 was a key target of Rbfox1 despite the many other targets was to use a BDNF induced LTP paradigm. As responses in this setup would be expected to be dominated by BDNF/TrkB signaling, any regulator of TRkB levels would be expected to have strong phenotype here and dominate over the impact of other genes. Perhaps another complimentary functional experiment, if possible would be helpful here.

The reviewer’s point is well taken and indeed there could be other genes that are regulated by RbFox1 and that could impact TrkB levels or signaling. To strengthen our observation and test the direct regulation of TrkB.T1 in BDNF-induced LTP, we have performed this experiment in the context of conditional TrkB.T1 deletion in neurons to clarify that the previous rescue effect was not caused by an indirect down-regulation of TrkB.T1 in glia (new Figure 8). Moreover, we have performed a new complementary functional experiment to show that increasing or decreasing TrkB.T1 modulates TrkB.FL signaling, which is key to BDNF-induced LTP (New Figure 6).

4) Related to the above, it could be interesting to determine whether any of the other RNA-seq hits were TrekB dependent, this could be another opportunity to support a key role for Rbfox1 dependent regulation of TrkB.T1 in downstream phenotypes.

We agree that it would be nice to know if there are some TrkB- dependent RNA-seq hits. However, we are afraid that it would be almost impossible to demonstrate a direct relationship between those hits and TrkB-induced LTP because there are so many genes/neuronal pathways that are dysregulated by Rb-Fox1. To convey this point, we have changed panel Figure 5A to indicate that many neuronal pathways are dysregulated. Yet, we hope that the reviewer will agree that it is remarkable that correcting only the expression of one isoform can rescue the BDNF-induced LTP.

5) The authors comment on potential differences between mouse and human, and it might be helpful to confirm that Rbfox1 similarly regulates TrkB.T1 in human neurons and determine whether knockdown of Rbfox1 is effective in that system.

Following the reviewer’s comment we have investigated the possibility of testing our hypothesis in human hippocampal neurons, which are available. However, National Institutes of Health (NIH) regulations do not allow such experiments since human hippocampal neurons are of fetal origin. We have also considered the possibility of differentiating human iPSCs or ES cells but, to our knowledge, there are, as yet, no good protocols specific to the differentiation of hippocampal neurons. In the future, it will be important to test RbFox1 in conjunction with TrkB levels in patients with specific neurodegenerative or neuropsychiatric disorders, studies that we think will be prompted by our report.

[Editors' note: the author responses to the re-review follow.]

We are pleased to inform you that your article, "RbFox1 up-regulation impairs BDNF-dependent hippocampal LTP by dysregulating TrkB isoform expression levels", has been accepted for publication in eLife. A summary of the study and the reviewers' comments are included below.The BDNF TrkB receptor is responsible for synaptic plasticity, obesity and learning and memory. In addition to the full length TrkB tyrosine kinase receptor, a truncated form of TrkB (TrkB.T1) exists in the central nervous system. However the regulation of TrkB.T1 receptors is not well understood. In this study, the RNA binding protein, RBFox1, was found to upregulate the expression of TrkB.T1, which resulted in a deficit in BDNF-dependent LTP. This paper provides a new mechanism to explain the effects of BDNF upon synaptic plasticity.The revised manuscript has been evaluated by two prior reviewers. Both reviewers found the revised manuscript adequately addressed the previous criticisms and questions with additional experiments and analysis. They concluded the revision is suitable for publication. One remaining question was whether the regulation of truncated TrkB occurred through RNA stability or alternatively by a splicing event. this issue can be dealt with in the future, but it would be useful to acknowledge it in the text of the paper.

We have included the modification requested by the reviewers and indicated that future experiments will be needed to address the mechanism by which TrkB.T1 is regulated. The paragraph with the requested modifications has been included in the Discussion.